# W-Edit: A Wavelet-Based Frequency-Aware Framework for Text-Driven Image Editing

**Jiahui Sun**[1,2], **Weining Wang**[2*], **Mingzhen Sun**[2], **Peiyao Wang**[2], **Xinxin Zhu**[2], **Jing Liu**[1,2]

[1]School of Advanced Interdisciplinary Sciences, University of Chinese Academy of Sciences
[2]Institute of Automation, Chinese Academy of Sciences
`sunjiahui2023@ia.ac.cn`, `weining.wang@nlpr.ia.ac.cn`

## Abstract

While recent advances in Diffusion Transformers (DiTs) have significantly advanced text-to-image generation, text-driven image editing remains challenging. Existing approaches either struggle to balance structural preservation with flexible modifications or require costly fine-tuning of large models. To address this, We introduce W-Edit, a training-free framework for text-driven image editing based on wavelet-based frequency-aware feature decomposition. W-Edit employs wavelet transforms to decompose diffusion features into multi-scale frequency bands, disentangling structural anchors from editable details. A lightweight replacement module selectively injects these components into pretrained models, while an inversion-based frequency modulation strategy refines sampling trajectories using structural cues from attention features. Extensive experiments demonstrate that W-Edit achieves high-quality results across a wide range of editing scenarios, outperforming previous training-free approaches. Our method establishes frequency-based modulation as both a sound and efficient solution for controllable image editing.

## 1 Introduction

In recent years, diffusion models have become the dominant paradigm for image generation (Ho et al., 2020; Sohl-Dickstein et al., 2015; Song et al., 2020; Song & Ermon, 2019), achieving remarkable progress in large-scale text-to-image (T2I) generation. Early designs such as Stable Diffusion (Rombach et al., 2022) employ U-Net (Ronneberger et al., 2015) backbones, whereas recent frameworks including FLUX (Labs, 2024) and SD3 (Esser et al., 2024) replace U-Net with Diffusion Transformers (DiT) (Peebles & Xie, 2023) and integrate flow matching (Albergo & Vanden-Eijnden, 2022; Lipman et al., 2022; Liu et al., 2022) to improve fidelity and efficiency. Despite these advances in generation, controllable text-driven image editing remains a significant challenge.

Text-driven image editing aims to modify images according to natural language instructions, supporting operations such as object replacement, attribute modification, style transfer, and background editing, as shown in Fig. 1. The central difficulty lies in achieving precise and flexible modifications while preserving the global layout and contextual coherence of the reference image. Existing methods fall into two groups. Training-based methods (Zhang et al., 2023b; Kawar et al., 2023; Zhang et al., 2023a) adapt diffusion models to reconstruct reference images before applying edits. While effective, this process is computationally expensive and prone to catastrophic forgetting. In contrast, training-free approaches achieve editing by modifying the sampling trajectory of pretrained models at inference time, without additional training. Among them, Inversion-based methods (Brack et al., 2024; Cao et al., 2023; Deutch et al., 2024; Aberman et al., 2024) avoid retraining by mapping images into noise space and resampling with new prompts, but suffer from trajectory drift and limited controllability. Attention-injection approaches such as Prompt-to-Prompt (Hertz et al., 2022) improve structure preservation but are highly sensitive to layer selection and fail under complex edits. More recently, Stable-Flow (Avrahami et al., 2025b) introduced the idea of vital layer in flow-based diffusion models and performs injection only at these selected layers. This strategy improves stability and enables the completion of certain edits, but it enforces overly rigid constraints, often

---

*Corresponding author.

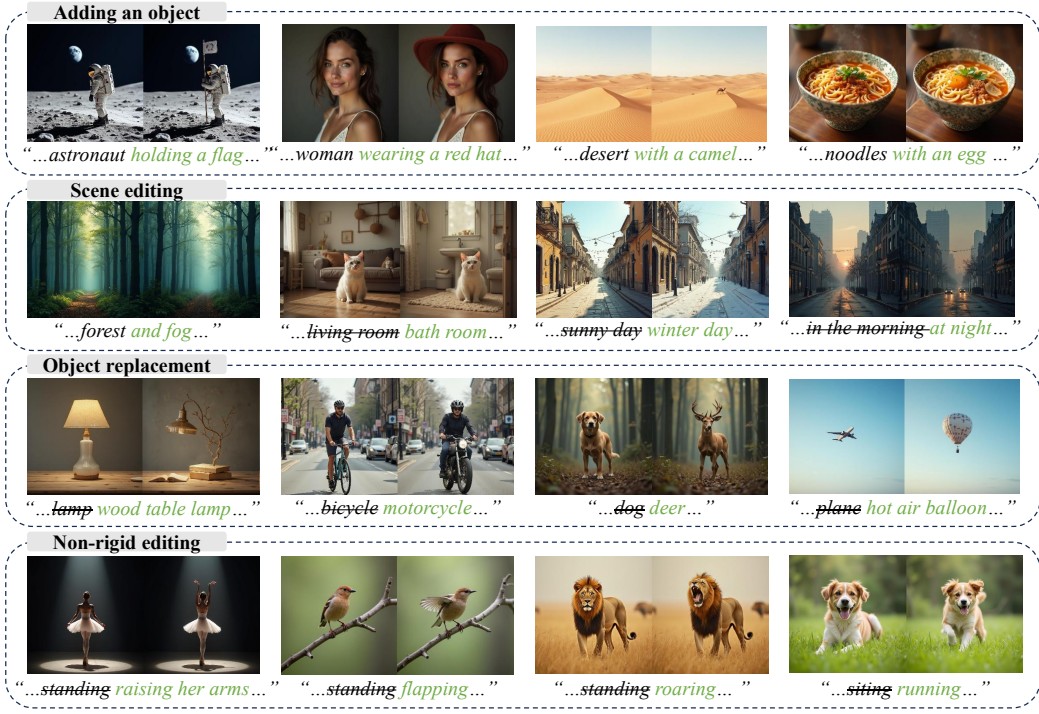

Figure 1: **Representative editing results produced by W-Edit.** Our training-free editing method is able to perform various types of image editing operations, including non-rigid editing, object addition, object removal, and global scene editing. These different edits are accomplished using the same mechanism.

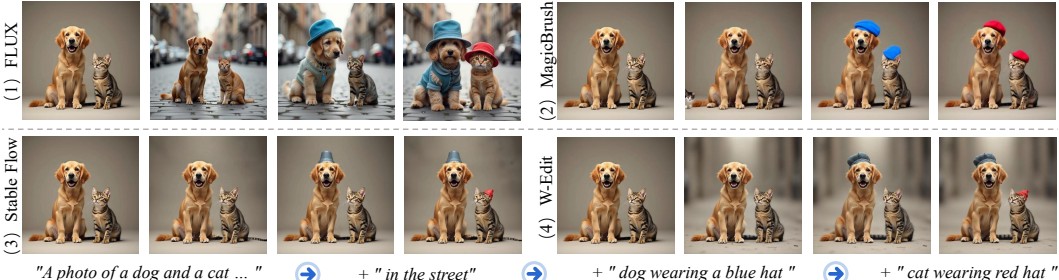

Figure 2: **Comparison of editing consistency and modification fidelity across representative methods.** Using different textual prompts with the same initial seed, we observe the following: (1) FLUX produces relatively stable outputs but often fails to accomplish the intended edits; (2) MagicBrush struggles to generate precise edits that faithfully follow textual instructions; (3) Stable-Flow delivers more stable results and completes editing tasks, but its mechanism enforces overly rigid constraints on diversity and can still introduce inconsistencies (e.g., the background remains unchanged despite text modifications); and (4) in contrast, W-Edit produces edits that align closely with textual descriptions while preserving the consistency of unrelated content. Fine details are best viewed when zoomed in.

preventing necessary changes such as scene-level modifications. As shown in Fig. 2, these methods either maintain structure while missing edits, or realize edits at the cost of consistency.

We argue that this difficulty arises from the entanglement of global semantics (e.g., layout, object identity) and local signals (e.g., texture, color, fine attributes) in the spatial domain. This entanglement makes it difficult to simultaneously preserve structure and introduce modifications. The frequency domain, in contrast, offers a natural decomposition that aligns with editing objectives. Low-frequency components encode layouts and semantics and thus can serve as reliable anchors for consistency, while high-frequency components capture textures and variations and thus are suitable

for flexible modification. This natural separation aligns directly with editing objectives, enabling explicit control over which parts of the image should remain stable and which parts can be modified. Building on this insight, we analyze the feature representations of Diffusion Transformers (DiTs) from a frequency-domain perspective and discover a block-wise frequency progression. We find that early blocks predominantly focus on low-frequency structures, while later blocks refine high-frequency details as shown in Fig. 4. Building on this insight, we reformulate text-driven editing as multi-level frequency control.

Building on this progression, we propose **W-Edit**, a training-free framework for text-guided image editing from a frequency-domain perspective. We introduce three key innovations. First, we employ wavelet-guided decomposition to provide multi-scale frequency bands with spatial locality, allowing separate treatment of structure and detail. Second, we design a lightweight frequency-band replacement module that selectively injects these components into pretrained diffusion models, avoiding the need for costly retraining. Third, we develop an inversion-based frequency modulation strategy that refines the sampling trajectory using structural cues from attention features, minimizing structural drift and improving controllability.

Extensive experiments show that W-Edit achieves high-quality results across diverse editing scenarios. Compared with prior training-free approaches, W-Edit delivers more faithful, consistent, and visually coherent edits, establishing frequency-based modulation as a principled and efficient solution for controllable diffusion editing.

## 2 RELATED WORK

### 2.1 TEXT-GUIDED IMAGE EDITING METHODS

Text-driven image editing usually builds upon pretrained T2I diffusion models (Huberman-Spiegelglas et al., 2024; Samuel et al., 2023; Tumanyan et al., 2023; Wallace et al., 2023; Wu & De la Torre, 2023), ranging from early U-Net based backbones such as Stable Diffusion to recent Transformer based models like FLUX (Labs, 2024) and SD3 (Esser et al., 2024). Existing approaches can be divided into training-based and training-free methods. Training-based methods, such as InstructPix2Pix (Brooks et al., 2023) and MagicBrush (Zhang et al., 2023a), construct large-scale instruction–image triplets to learn direct mappings for editing. IMagic (Kawar et al., 2023) and SINE (Zhang et al., 2023b) reconstruct reference images by adapting pretrained models, but this is costly and prone to catastrophic forgetting. While effective, these methods demand heavy annotation and training costs and often generalize poorly to unseen domains such as video or fine-grained edits. Training-free methods (Brack et al., 2024; Cao et al., 2023; Deutch et al., 2024; Aberman et al., 2024), which dominate recent research, avoid dataset construction and leverage pretrained priors instead. Among them, inversion-based approaches (Song et al., 2020; Wu & De la Torre, 2023) reconstruct a reference image in the diffusion latent space before applying edits, but suffer from trajectory drift or reconstruction errors. Later improvements, such as optimization-based calibration and structural regularization (Brack et al., 2024; Cao et al., 2023; Aberman et al., 2024; Parmar et al., 2023; Tumanyan et al., 2023), enhance stability and control. Attention-modulation techniques (Hertz et al., 2022; Tumanyan et al., 2023) improve spatial control by injecting cross- and self-attention features, yet they remain sensitive to block selection, computationally expensive, and prone to semantic inconsistency. While these methods achieve promising results, they still struggle to preserve global structure while enabling flexible local modifications without retraining.

### 2.2 FREQUENCY PERSPECTIVE IN DEEP LEARNING

Beyond spatial-domain manipulation, frequency-domain analysis has proven useful in a variety of vision tasks. Early CNN-based studies integrated Discrete Cosine Transform (DCT) to accelerate convergence (Ghosh & Chellappa, 2016) or introduced frequency-aware dynamic networks for super-resolution (Xie et al., 2021). Though not intended for editing, these studies demonstrated the potential of frequency representations in vision. More recently, diffusion-based works further leverage frequency cues for generation and editing: FDDiff (Wang et al., 2024) restores high-frequency details in low-resolution images, FDG-Diff (Zhang et al., 2025) enforces spatial–frequency consistency, WaveDiff (Phung et al., 2023) decomposes latent features into sub-bands for improved texture preservation, FreeU (Si et al., 2024) regulates frequency responses to improve the generation qual-

ity, ConsisID (Yuan et al., 2025) maintains identity in video generation via frequency control, and FCDiffusion (Gao et al., 2024) designs multi-branch frequency pathways for diverse translations. However, these methods typically require additional objectives or architectural modifications, limiting their plug-and-play applicability. This line of inquiry has been extended to image editing, where methods like FlexiEdit (Koo et al., 2024) refine the DDIM latent space by selectively attenuating high-frequency components to enhance fidelity in non-rigid edits, and FDS (Ren et al., 2025) proposes a frequency-aware denoising score to improve the stability and quality of text-guided edits. However, these methods typically require additional objectives or architectural modifications, limiting their plug-and-play applicability. In contrast, W-Edit introduces a training-free, wavelet-based frequency modulation framework for text-driven image editing. Instead of relying on handcrafted attention injections or heavy retraining, it decomposes intermediate diffusion features into multi-scale wavelet bands and selectively injects them to guide the sampling process. This lightweight design enables controllable and structurally consistent editing across diverse tasks, bridging the gap between frequency-domain analysis and practical editing applications.

## 3 METHOD

Our goal is to enable controllable text-driven image editing without retraining preserves consistency while producing faithful modifications. To this end, we introduce W-Edit, a wavelet-based framework that disentangles global structure and local details in the frequency domain. As illustrated in Fig. 3, we first extract an inversion trajectory from the reference image, then applies wavelet-based frequency decomposition and fusion at selected blocks, and finally performs frequency-guided editing with structural cues. Specifically, Sec. 3.1 introduces wavelet-based frequency decomposition, Sec. 3.2 analyzes block-level frequency characteristics that underpin structural and detail preservation, and Sec. 3.3 details our wavelet-based editing framework and modulation strategy.

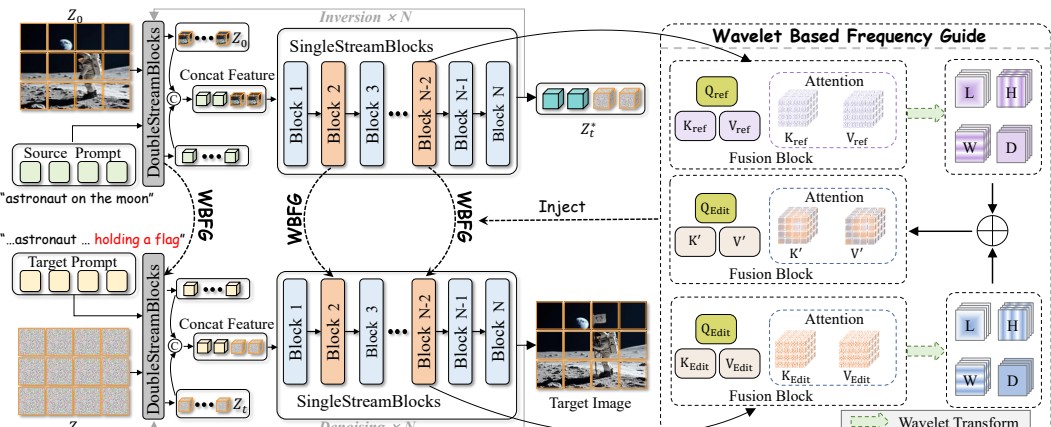

Figure 3: **Overall Framework of W-Edit.** Given an input image and a corresponding text instruction, we first perform inversion on the input image to obtain the initial noise and the inversion trajectory. Next, starting from both the inverted noise and new random noise, we sample to generate the reference image and the edited image simultaneously. Within the pretrained T2I model, we selectively choose different Transformer blocks and decompose the reference image's keys $(K)$ and values $(V)$ into multi-scale frequency bands using wavelet transforms. An energy-aware adaptive frequency fusion mechanism is then applied to inject these frequency-domain features from the reference image into the sampling process of the edited image. This procedure guides the execution of the editing task, resulting in high-quality images that preserve structural consistency while accurately reflecting the desired semantic modifications.

### 3.1 WAVELET-BASED FREQUENCY DECOMPOSITION

A central challenge in editing is that global semantics (e.g., layout, object identity) and local details (e.g., texture, color, fine attributes) tend to be entangled in the spatial domain. This entanglement makes it difficult to preserve unchanged components while selectively modifying target regions. To

address this issue, we seek to project diffusion features into the frequency domain, where structural anchors and fine details can be explicitly disentangled. Traditional transforms like the Fourier Transform provide a global frequency decomposition but discard spatial localization, making them unsuitable for tasks that require local or multi-scale control. To overcome this limitation, we adopt the wavelet transform, which introduces basis functions localized in both space and frequency, enabling to simultaneously capture global structures and fine details in a principled way.

Suppose that $t$ represents an independent variable. A mother wavelet $\psi(t)$ generates scaled and shifted versions $\psi_{a,b}(t)$, where $a$ is the *scale parameter* controlling dilation (larger $a$ corresponds to lower frequencies), and $b$ is the *translation parameter* controlling position along the $t$-axis. The continuous wavelet transform (CWT) provides a joint time-frequency representation $W(a,b)$:

$$\psi_{a,b}(t) = \frac{1}{\sqrt{a}}\psi\left(\frac{t-b}{a}\right), \quad a > 0, \quad b \in \mathbb{R}, \qquad W(a,b) = \int_{-\infty}^{+\infty} f(t)\psi_{a,b}^*(t)dt. \tag{1}$$

The discrete wavelet transform (DWT), in contrast, yields a compact hierarchical decomposition. For 2D images, one-level DWT produces four sub-bands:

$$\mathbf{F} \xrightarrow{\text{DWT}} \mathbf{F}_A, \mathbf{F}_H, \mathbf{F}_V, \mathbf{F}_D, \tag{2}$$

where $\mathbf{F}_A$ is the low-frequency approximation, and $\mathbf{F}_H, \mathbf{F}_V, \mathbf{F}_D$ capture horizontal, vertical, and diagonal high-frequency details. Recursive application of DWT to $\mathbf{F}_A$ yields a multi-level decomposition that represents both global structures and fine-grained textures. This decomposition naturally separates global and local signals: low-frequency components encode layout and semantics, serving as reliable anchors for structural consistency, while high-frequency components capture textures and variations, enabling flexible and precise modifications. Such multi-scale, spatially localized frequency representation forms the foundation of our W-Edit framework.

This decomposition provides a natural separation of consistency (low-frequency) and editability (high-frequency), forming the basis for frequency-aware analysis in Sec. 3.2.

## 3.2 BLOCK-WISE FREQUENCY PROGRESSION IN DIFFUSION TRANSFORMERS

In U-Net based diffusion models, editing is often heuristically guided by selecting early blocks for structure and later blocks for details. This intuition stems from the encoder–decoder hierarchy of U-Nets, where spatial resolution decreases and then recovers. However, such heuristics do not apply to Diffusion Transformers (DiTs), whose blocks are architecturally homogeneous and lack clear semantic stages. As a result, it is unclear how information is organized across blocks, and whether some blocks emphasize structural cues while others capture fine details.

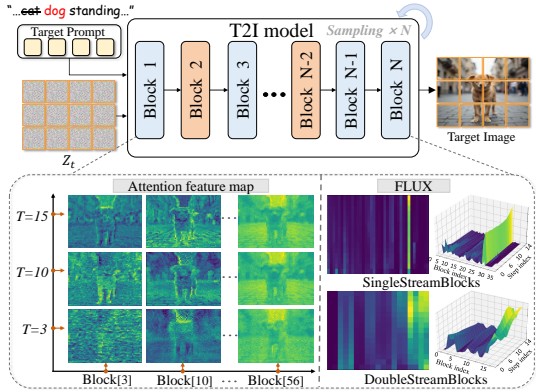

Figure 4: **Frequency progression in T2I models.** Attention features in DiTs progressively capture coarse layouts in early blocks and refined details in later blocks, with outputs progressively refined during sampling, providing strong image priors for generation.

To address this question, we analyze intermediate features of DiTs in the frequency domain. Suppose that $z^k$ is the output feature map at the $k$-th block, where each block processes the intermediate feature at a specific stage of the diffusion process. We compute the Fourier transform of $z^k$ as:

$$\hat{z}^k(u,v) = \mathcal{F}[z^k], \quad (u,v) \in \mathbb{F}^2 \tag{3}$$

where $\hat{z}^k(u,v)$ denotes the frequency-domain representation of $z^k$, $(u,v)$ are horizontal and vertical frequency indices and $\mathcal{F}\{\cdot\}$ denotes the 2D discrete Fourier transform, and $\mathbb{F}^2$ represents the 2D frequency domain.

To quantify frequency contributions, we define a mid-to-high frequency energy metric. We first define the Mid-Frequency Radius, $r_{\text{mid}}$, which acts as the boundary separating the low-frequency and high-frequency regimes

$$r_{\text{mid}} = r_{\text{max}}//2 \tag{4}$$

where $r_{\max}$ is the maximum possible radial distance in the frequency domain. This $r_{\mathrm{mid}}$ specifies the radius corresponding to the intermediate frequency band and serves as our division threshold. We then define the Mid-To-High Frequency Energy ($E^k_{\mathrm{MTH}}$) as the total energy accumulated from the radius $r_{\mathrm{mid}}$ up to the maximum frequency:

$$E^k_{\mathrm{MTH}} = \sum_{r=r_{\mathrm{mid}}}^{\mathrm{max\_radius}} \mathcal{P}_r(\hat{z}^k) \tag{5}$$

Here, $\mathcal{P}_r(\hat{z}^k)$ is the radial power spectrum (the average energy at radius $r$). $E^k_{\mathrm{MTH}}$ is a scalar value that directly measures the mid-to-high frequency components of the feature map.

For completeness, low- and high-frequency energies are computed as:

$$E^k_{\mathrm{low}} = \sum_{(u,v)\in\Omega_{\mathrm{low}}} |\hat{z}^k(u,v)|^2, \qquad\qquad E^k_{\mathrm{high}} = \sum_{(u,v)\in\Omega_{\mathrm{high}}} |\hat{z}^k(u,v)|^2 \tag{6}$$

Here, $\Omega_{\mathrm{low}}$ and $\Omega_{\mathrm{high}}$ denote frequency index sets determined by radial thresholds. We compute the $E^k_{\mathrm{MTH}}$ for both SingleStreamBlocks and DoubleStreamBlocks of FLUX and visualize the results together with the attention maps in Fig. 4. Since these blocks employ different fusion strategies, we visualize them separately to highlight distinct frequency behaviors. In the visualizations, darker regions correspond to low-frequency components, while brighter regions indicate higher frequencies. The contours in the attention maps show that early blocks mainly encode low-frequency structural foundations, whereas later blocks exhibit sparser attention distributions that refine high-frequency details. Based on these observations, we select blocks with either very high or very low mid-to-high frequency energy as the target blocks for frequency-domain information fusion.

To validate this choice, we compare our selected blocks with the vital layers identified in prior work (Avrahami et al., 2025a), in which blocks importance was measured using DINOv2 (Oquab et al., 2023). We find that these vital layers largely coincide with our selected fusion blocks, indicating a strong correspondence between block importance and frequency response. Leveraging these functional characteristics, we implement selective frequency-domain injection across different modules. This strategy offers two main advantages. First, it allows each module to operate optimally within its functional domain. Second, it enables frequency-aware guidance during sampling, significantly reducing computational overhead while maintaining precise control over both structure and fine-grained details.

### 3.3 WAVELET-BASED FREQUENCY-GUIDED EDITING

As illustrated in Fig. 3, our method builds upon pretrained diffusion or flow models. We employ wavelet-based frequency modulation to selectively replace specific frequency bands. This enables precise, flexible, and high-fidelity visual editing in pretrained diffusion models. Notably, our approach is training-free and compatible with a wide range of mainstream diffusion architectures. The framework comprises two primary trajectories: the *inversion trajectory* and the *editing trajectory*. During inversion, intermediate noisy latents and key–value $(K, V)$ attention pairs are recorded, capturing the evolutionary trajectory of the source image and serving as references for subsequent editing. The editing trajectory starts from pure noise and incorporates these stored key–value pairs while following new textual instructions. This ensures that latent representations are guided toward the original structure, preserving maximum fidelity and editing precision.

We first implement our framework on FLUX, a flow model. Flow models generate samples by transporting a prior distribution $p_0$ (Gaussian noise) toward the data distribution $p_1$ (image manifold). In $\mathbb{R}^d$, we define a time-dependent density $p_t : [0,1] \times \mathbb{R}^d \to \mathbb{R}_{>0}$ and a vector field $u_t : [0,1] \times \mathbb{R}^d \to \mathbb{R}^d$. The flow $\phi_t$ is generated via the ODE:

$$\frac{d\phi_t(x)}{dt} = u_t(\phi_t(x)), \quad \phi_0(x) = x \tag{7}$$

transporting samples from $p_0$ toward $p_1$. To edit real images, we invert them into the latent space, i.e., transport samples from $p_1$ back to $p_0$. Using the reverse Euler solver in FLUX, given the forward update and its reverse:

$$z_{t-1} = z_t + (\sigma_{t+1} - \sigma_t)\, u_t(z_t), \qquad\qquad z_t = z_{t-1} + (\sigma_t - \sigma_{t+1})\, u_t(z_{t-1}) \tag{8}$$

where $z_t$ is the latent at timestep $t$, $\sigma_t$ is the transport standard deviation, and $u_t$ is the vector field, assuming $u_t(z_t) \approx u_t(z_{t-1})$ for sufficiently small steps. This inversion maps images back to noise while storing key–value pairs for editing.

**Frequency Decomposition via Wavelet Transform.** To capture multi-scale visual information, we apply discrete wavelet transform (DWT) to intermediate model features. Given a feature tensor $\mathbf{F} \in \mathbb{R}^{B \times C \times H \times W}$ (or flattened attention $\mathbf{F} \in \mathbb{R}^{B \times N \times D}$), we apply DWT to decompose it into low-frequency approximation $\mathbf{F}_A$ and high-frequency detail coefficients $\mathbf{F}_{D_i}$:

$$\mathbf{F} \xrightarrow{\text{DWT}} \{\mathbf{F}_A, \mathbf{F}_{D1}, \mathbf{F}_{D2}, \ldots, \mathbf{F}_{Dn}\} \tag{9}$$

where $\mathbf{F}_A$ encodes global structure, and $\mathbf{F}_{D_i}$ capture edges and textures. This provides a multi-scale representation for separate treatment of global structure and fine details.

**Adaptive Frequency Fusion.** We propose an **energy-aware adaptive frequency fusion** mechanism. Let $E_i = \sum |\mathbf{F}_{\text{ref},i}|^2$ denote the energy of the $i$-th sub-band. We select the minimal set of sub-bands whose cumulative energy reaches a threshold $\eta$:

$$\mathbf{F}'_i = \begin{cases} \mathbf{F}_{\text{ref},i}, & \text{if } \sum_{j=1}^i E_j \leq \eta \sum_k E_k, \\ \mathbf{F}_i, & \text{otherwise.} \end{cases} \tag{10}$$

Since visual energy in natural images is predominantly concentrated in low-frequency components representing global structure (Khayam, 2003; Pimpalkhute et al., 2021), this energy-based selection explicitly locks the reference image's layout. By preserving these high-energy bands and allowing the model to generate the remaining low-energy high-frequency bands, we achieve a balance where the scene structure remains consistent while fine-grained details are free to adapt to text instructions. $\eta$ balances guidance strength and feature preservation.

**Integration with Attention.** After fusion, coefficients are reconstructed via inverse DWT (IDWT) and then integrated into attention:

$$\mathbf{F}' \xleftarrow{\text{IDWT}} \{\mathbf{F}'_A, \mathbf{F}'_{D1}, \ldots, \mathbf{F}'_{Dn}\} \quad \Rightarrow \quad \text{Attn}'(\mathbf{Q}, \mathbf{K}', \mathbf{V}') = \text{Softmax}\left(\frac{\mathbf{Q}\mathbf{K}'^{\top}}{\sqrt{d}}\right)\mathbf{V}' \tag{11}$$

with $\mathbf{K}' = \mathbf{F}'W_K$, $\mathbf{V}' = \mathbf{F}'W_V$, and $\mathbf{Q} = \mathbf{F}W_Q$. Modulating key–value pairs in the frequency domain guides both global and local generation. Low-frequency bands control composition; high-frequency bands refine details. Energy-aware fusion allows blending multiple references or edits dynamically. Compared with DCT-based or learned frequency-control methods, our wavelet-based approach automatically selects informative frequency components according to energy distribution, offering superior spatial-frequency localization. Integration into attention enables precise, dynamic control while remaining training-free.

## 4 EXPERIMENTS

In Sec. 4.1, we benchmark W-Edit against baseline methods using both qualitative and quantitative evaluations. Sec. 4.2 presents a user study evaluating alignment, consistency, realism, and plausibility from a human perspective. Sec. 4.3 reports ablation studies that isolate the contribution of each component, while Sec. 4.4 demonstrates W-Edit's generality and adaptability across different diffusion architectures.

### 4.1 QUALITATIVE AND QUANTITATIVE COMPARISON

In this section, we compare the performance of W-Edit with several state-of-the-art text-driven image editing methods. Our framework is built upon the FLUX.1-dev model, and we adopt the official public implementations of P2P (Hertz et al., 2022), MagicBrush (Zhang et al., 2023a), Flow-Edit (Kulikov et al., 2024), and Stable-Flow (Avrahami et al., 2025b) for fair comparison.

**Qualitative Comparison.** As shown in Fig. 5, our proposed W-Edit achieves precise editing in the target regions while faithfully preserving the unedited areas. Specifically, existing methods often struggle to preserve the overall layout when applying local edits. For instance, Stable-Flow

distorts the bicycle pedals and other baselines misplace the basket, leading to global inconsistencies. In contrast, our method faithfully maintains the bicycle's frame geometry and basket position, demonstrating superior structural consistency. Similarly, for a snake coiled around a tree, Stable-Flow captures the pose but produces a physically implausible floating body and an anomalous tail, whereas W-Edit generates a natural configuration consistent with the scene's overall structure These examples demonstrate that W-Edit, by leveraging frequency-based modulation, can effectively balance local edits with global structure preservation, producing more realistic and consistent results compared to other methods.

**Quantitative Comparison.** In addition to qualitative comparisons, we quantitatively evaluate the methods using the PIE-Bench dataset (Ju et al., 2023), which is specifically constructed for the systematic validation of image editing methods. As shown in Table 1, W-Edit achieves the lowest FID of 65.44 and the highest CLIP score of 31.84 among all compared methods. This represents an 18.6% reduction in FID and a 4.5% improvement in CLIP score over Flow-Edit. These gains highlight its superiority in both visual realism and semantic alignment with the editing instructions, driven by advanced generative priors and our wavelet-based editing strategy that faithfully translating textual modifications while avoiding artifacts. Moreover, W-Edit consistently outperforms baselines by achieving a PSNR of 24.06, which is 14.5% higher than the second-best method, and a LPIPS of 0.1028, representing 32.5% improvement, underscoring its ability to maintain structural fidelity and perceptual quality. In particular, our method demonstrates a strong capability to preserve global composition and background details, while applying precise and localized edits in the target regions. Together, these results provide compelling evidence that W-Edit achieves a more favorable balance between fidelity, alignment, and controllability compared to existing training-free approaches. Moreover, compared with FLUX, our method only increases inference time by 10.8% and GPU memory usage by 1.6%. More efficiency comparisons are presented in Appendix A.2.

**VLM Evaluation.** In addition to conventional pixel or feature-level metrics, we employ vision–language models (VLMs) to provide a more semantic evaluation of editing quality. Following Stable-Flow, we use *Phi-3.5-vision* (Marah Abdin, 2024) to evaluate two criteria: **Text Following**, which measures whether the edited image matches the prompt, and **Minimal Modification**, which assesses whether edits are restricted to the described regions. As shown in Table 1, W-Edit achieves the best balance between semantic fidelity and structural preservation. Unlike MagicBrush, which tends to prioritize text alignment at the expense of global consistency, W-Edit faithfully follows instructions while avoiding unnecessary alterations in unedited areas.

Table 1: Quantitative Comparison of Different Methods

| Method | PIE-Bench | | | | VLM Evaluation | | User Study Scores | | | |
|---|---|---|---|---|---|---|---|---|---|---|
| | CLIP↑ | FID↓ | PSNR↑ | LPIPS↓ | Text Fol. | Modify | Align. | Consist. | Real. | Plaus. |
| P2P | 28.13 | 320.65 | 15.12 | 0.4736 | 31.5% | 24.0% | 2.7 | 3.0 | 2.9 | 2.6 |
| MagicBrush | 29.06 | 206.19 | 15.68 | 0.4615 | **84.5%** | 33.5% | 3.5 | 2.3 | 2.4 | 2.2 |
| Flow-Edit | 30.48 | 80.35 | 18.33 | 0.2642 | 76.0% | 54.5% | 3.6 | 3.6 | 3.6 | 3.4 |
| Stable-Flow | 29.16 | 89.78 | 21.02 | 0.1522 | 77.5% | 58.0% | 3.7 | 3.7 | 4.1 | 3.4 |
| W-Edit | **31.84** | **65.44** | **24.06** | **0.1028** | 81.0% | **63.0%** | **3.8** | **3.9** | **4.2** | **3.8** |

## 4.2 USER STUDY

We conducted a user study to evaluate our editing approach. Specifically, we used GPT-4 to generate 100 prompt sets, each consisting of four editing instructions covering object addition, removal, replacement, attribute modification, style transfer, and background change. Fifteen participants rated the editing results on a five-point Likert scale across four dimensions: (1) alignment with the target prompt, (2) consistency with the reference image, (3) realism, and (4) physical plausibility. As shown in Table 1, our method consistently outperforms all other methods across nearly all aspects, with particularly notable improvements in realism and physical plausibility.

## 4.3 ABLATION STUDY

To better understand the contribution of different components in W-Edit, we conduct an ablation study and report results in Tab. 2.

Among all configurations, selected-block injection achieves the best overall balance across the three CLIP-based metrics, yielding the highest average score of 45.48 (see Appendix A.1 for metric def-

| Input | P2P | MagicBrush | Flow-Edit | Stable-Flow | **W-Edit(ours)** |
|---|---|---|---|---|---|

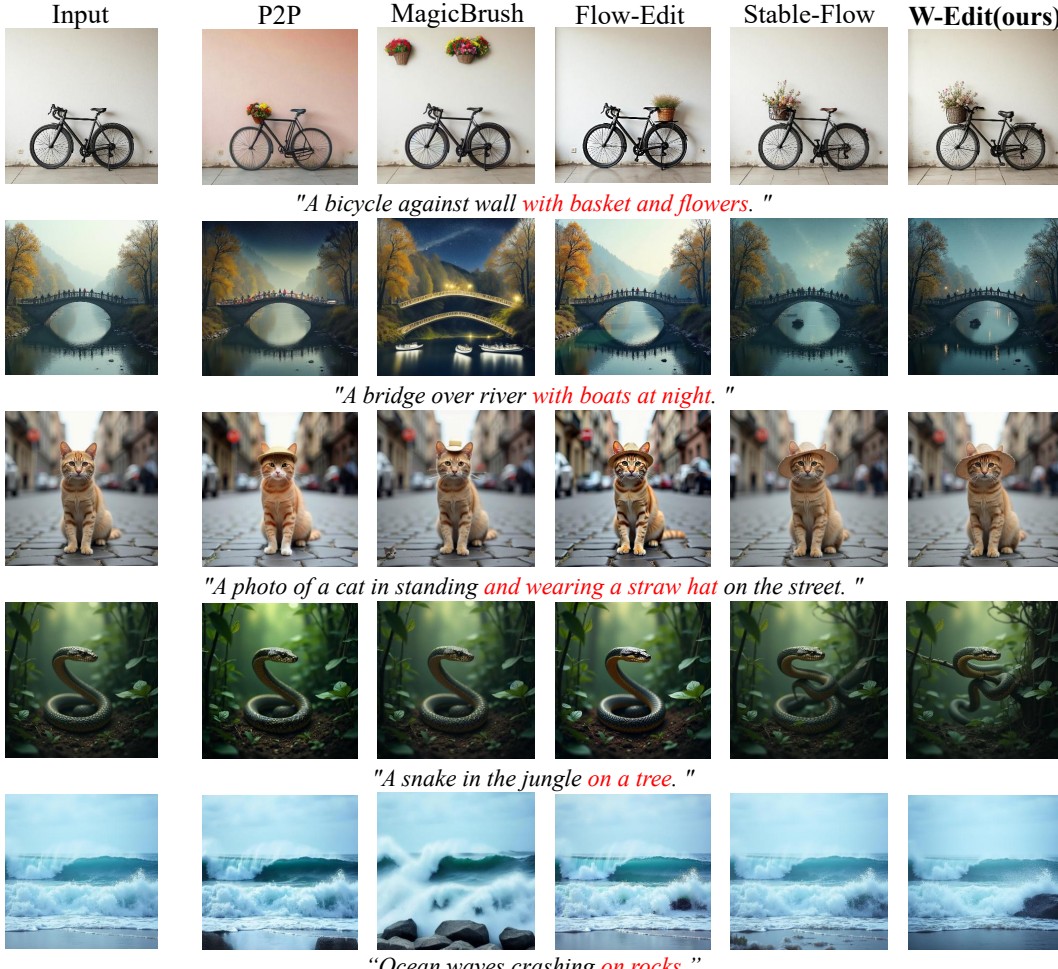

*"A bicycle against wall with basket and flowers. "*

*"A bridge over river with boats at night. "*

*"A photo of a cat in standing and wearing a straw hat on the street. "*

*"A snake in the jungle on a tree. "*

*"Ocean waves crashing on rocks."*

Figure 5: **Qualitative comparison.** Our method more faithfully adheres to target prompts while preserving the original image's structure.

initions). In contrast, all-block injection maximizes image similarity but almost collapses in directional consistency, suggesting that overly aggressive injection enforces the reference too strongly and suppresses textual guidance. Removing SingleStreamBlocks or DualStreamBlocks also leads to performance degradation. The absence of SingleStreamBlocks harms similarity, while excluding DualStreamBlocks reduces text alignment and consistency. Frequency components further play complementary roles. Without high-frequency signals, the model loses fine-grained consistency, whereas ablating low-frequency components damages structural preservation, resulting in lower CLIPimg (See Appendix A.2 for visualization of frequency domain component decoupling.). These findings confirm that W-Edit benefits from both frequency bands and carefully chosen fusion blocks, which together enable a favorable trade-off between fidelity, alignment, and editing controllability.

To determine the optimal value for $\eta$, we performed a quantitative sensitivity analysis on the PIE-Bench dataset1. We evaluated performance using three complementary CLIP-based metrics. As shown in the Tab 3, $\eta$ functions as a critical slider balancing structural preservation (CLIPimg) and editability (CLIPtxt/CLIPdir). Lower values ($\eta < 0.4$) favor text alignment

Table 2: Ablation study

| Method | CLIPimg | CLIPtxt | CLIPdir | Average |
|---|---|---|---|---|
| Selected-block injection | 0.9749 | 0.3068 | 0.0826 | 0.4548 |
| All-block injection | 0.9988 | 0.2839 | 0.0013 | 0.4280 |
| w/o SingleStreamBlocks | 0.9184 | 0.3162 | 0.0880 | 0.4409 |
| w/o DualStreamBlocks | 0.9458 | 0.3089 | 0.0871 | 0.4473 |
| w/o high-frequency | 0.9391 | 0.3092 | 0.0821 | 0.4301 |
| w/o low-frequency | 0.9249 | 0.3125 | 0.0954 | 0.4443 |

at the cost of structural integrity, while higher values ($\eta > 0.8$) over-preserve the reference, suppressing the intended edit. Among the tested parameters, 0.6 achieved the highest CLIPdir score indicating the most accurate semantic transformation while maintaining a high degree of structural

fidelity. These quantitative findings align with the qualitative visualizations presented in Fig 6. Consequently, we fixed $\eta = \mathbf{0.6}$ to maintain robust performance across diverse tasks while ensuring framework simplicity and efficiency.

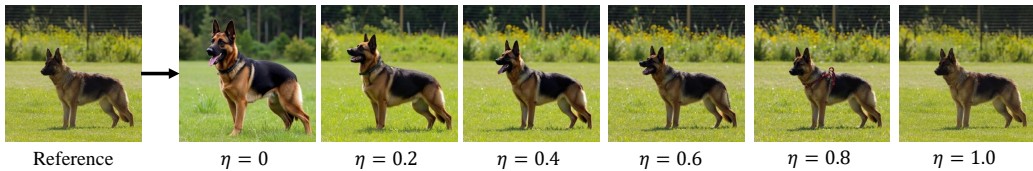

Reference     $\eta = 0$     $\eta = 0.2$     $\eta = 0.4$     $\eta = 0.6$     $\eta = 0.8$     $\eta = 1.0$

Figure 6: Visualization of the impact of different $\eta$ values on the generated results

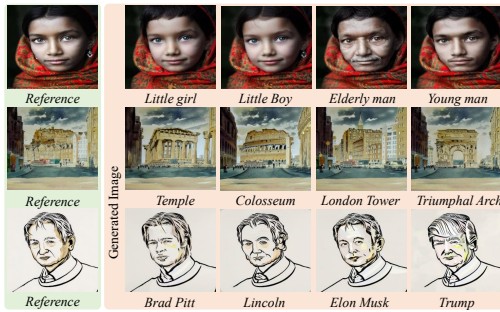

Figure 7: Qualitative editing results achieved on StableDiffusion-v1.5.

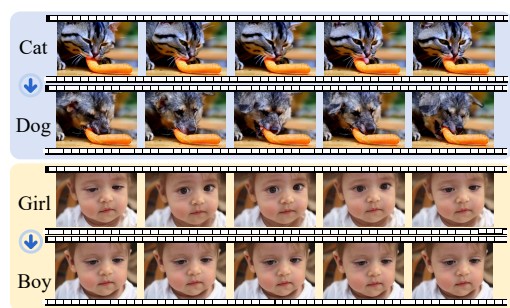

Figure 8: Qualitative editing results achieved on CogVideoX-1.0.

### 4.4 GENERALIZATION ANALYSIS

To evaluate the generalization ability of our method, we apply W-Edit to StableDiffusion-v1.5 and CogVideoX-1.0 (Yang et al., 2024), achieving effective style transfer and subject replacement (see Fig. 7 and Fig. 8). On StableDiffusion-v1.5, it transforms a girl into multiple identities, generates diverse subjects under consistent style, and converts images into architectural styles like London landmarks and the Arc de Triomphe. On CogVideoX-1.0, it performs coherent video edits, such as cat-to-dog and girl-to-boy transformations, maintaining background consistency and preserving identity cues.

Table 3: Sensitivity Analysis of ($\eta$)

| $\eta$ | CLIPimg (↑) | CLIPtxt (↑) | CLIPdir (↑) | Avg. (↑) |
|---|---|---|---|---|
| 0.2 | 0.8920 | 0.3210 | 0.0540 | 0.4223 |
| 0.4 | 0.9350 | 0.3140 | 0.0780 | 0.4423 |
| 0.6 | 0.9749 | 0.3068 | 0.0826 | 0.4548 |
| 0.8 | 0.9910 | 0.2760 | 0.0610 | 0.4427 |
| 1.0 | 0.9990 | 0.2620 | 0.0050 | 0.4220 |

## 5 CONCLUSIONS

We present W-Edit, a training-free image editing framework leveraging wavelet decomposition and frequency-guided modulation. By exploiting multi-scale frequency analysis, it ensures consistency and stability across editing operations. Extensive experiments show competitive performance and strong generality in tasks such as object addition, removal, and replacement. Our frequency-domain perspective offers a novel avenue for integrating spectral priors into editing and generative frameworks, paving the way for more controllable and versatile content creation.

## 6 ACKNOWLEDGEMENTS

This research is supported by Artificial Intelligence-National Science and Technology Major Project (2023ZD0121200), the National Natural Science Foundation of China (62531026, 62437001, 62436001), the Natural Science Foundation of Jiangsu Province under Grant BK20243051and the Strategic Priority Research Program of Chinese Academy of Sciences under Grant XDA04080400.

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

# A   APPENDIX

## A.1   EXPERIMENTAL EVALUATION SETUP

**Experimental Details.**   Given the advanced generative capabilities of FLUX, we first conduct experiments on the FLUX framework, where we select blocks [0, 1, 2, 17, 18, 28, 53, 54, 56] for frequency-domain fusion. Haar wavelets are used as the basis for multi-level decomposition, with the maximum frequency energy weight set to 0.6. For CogVideoX-1.0, we experiment with blocks [0, 1, 2, 3], employing the bior3.3 wavelet basis for decomposition. On Stable Diffusion v1.5, we operate on the overall features between UNet blocks, using Haar as the wavelet basis. To dispel concerns that the frequency variations shown in Fig 4 might be isolated cases and to verify the generalizability of our observations, we conducted the following extended experiments: We constructed a dataset containing $k = 64$ diverse text cues (generated by ChatGPT, covering different environments, object sets, and subjects) and calculated the average frequency energy distribution of these samples. The results strongly confirm that the transition from low-frequency structure dominance in early blocks to high-frequency detail dominance in later blocks is a consistent feature of the FLUX DiT architecture, rather than an artifact specific to any particular image. This statistical result strongly supports the robustness of our frequency-based block selection strategy.

**Datasets.**   PIE-Bench (Prompt-based Image Editing Benchmark) is a benchmark dataset specifically constructed for the systematic validation of image editing methods. It aims to evaluate the editing strategy proposed in this study and compare it with existing inversion methods, while addressing the lack of standardized evaluation criteria for current image inversion and editing techniques. The dataset comprises 700 images covering both natural scenes and artificial scenes (such as paintings), which are divided into ten editing types: random editing (written by volunteers), object modification, object addition, object deletion, object content alteration, object pose change, object color adjustment, object material modification, background replacement, and image style transformation. Within each editing type, the images are evenly distributed between natural and artificial scenes and balanced across four categories: animals, humans, indoor environments, and outdoor environments. Each image sample includes five key annotations: a source image prompt, a target image prompt, an editing instruction, a description of the editing subject, and an editing mask. The editing mask is particularly critical for accurately defining the expected editing region and is essential for precision evaluation. The dataset was constructed using a combination of automated and manual methods—random editing types were directly written by volunteers, while the other types utilized BLIP-2 to generate source image prompts and GPT-4 to produce target prompts and editing instructions. The results were then manually corrected to ensure accuracy, with two annotators and one reviewer collaboratively completing the annotation of editing subjects and masks. This process provides a standardized, high-quality evaluation benchmark for image editing research.

**VLMs Experiment(Phi-3.5-vision)**   Phi-3.5-vision is a highly capable VLM that demonstrates strong proficiency in both visual perception and linguistic reasoning. Its architecture is designed to enable joint understanding and reasoning about the complex interplay between visual content and textual descriptions. The utilization of the VLM for quantitative evaluation is motivated by several key factors. Firstly, traditional automated metrics, such as LPIPS or PSNR, primarily focus on low-level pixel-wise or structural similarity, which often fail to capture high-level semantic alignment between images and text prompts. In contrast, VLMs like Phi-3.5-vision are inherently suited for this task, as they can directly assess whether the visual content semantically conforms to the provided textual instruction. Secondly, evaluating "Minimal Modification" requires a model to not only understand the intended edit but also to recognize and preserve the unaltered regions of the source image. The joint understanding capability of VLMs allows them to perform this nuanced comparison more effectively than metrics that operate on image pairs without semantic guidance. By leveraging the reasoning capabilities of Phi-3.5-vision, our evaluation framework delivers a more semantically meaningful and reliable performance measure, particularly for instruction-based image editing tasks where faithfulness to both the text and the original image structure is paramount.

**CLIP Metrics**   To comprehensively evaluate the performance of our image editing framework, we employ three CLIP-based metrics that measure different aspects of editing quality as follows.

(1) CLIPimg (Image Similarity) quantifies the visual fidelity between the edited image and the original input by measuring their feature-space cosine similarity in the CLIP image encoder. Higher values (closer to 1.0) indicate better preservation of the original image's content and structure, ensuring that edits maintain the essential characteristics of the source material.

(2) CLIPtxt (Text Alignment) assesses how well the edited image corresponds to the target textual instruction by computing the cosine similarity between CLIP image features of the output and CLIP text features of the prompt. This metric captures the semantic alignment between visual content and textual description, with higher values indicating more accurate interpretation of editing commands.

(3) CLIPdir (Directional Consistency) measures the precision of attribute manipulation by calculating the cosine similarity between the vector from original to edited image in CLIP space and the vector from source to target text. This metric specifically evaluates whether the editing direction follows the intended semantic pathway, with positive values indicating consistent transformations and negative values suggesting deviation from the desired editing trajectory.

Together, these three metrics provide a multi-dimensional assessment of image editing systems, balancing the trade-offs between content preservation (CLIPimg), instruction faithfulness (CLIPtxt), and editing precision (CLIPdir).

## A.2 QUALITATIVE AND QUANTITATIVE EXPERIMENTS.

To comprehensively demonstrate W-Edit's versatility across diverse editing scenarios, we present qualitative results for six core editing tasks: **non-rigid editing**, **object addition**, **object removal**, **object replacement**, **attribute modification**, and **background editing**. Each task is designed to evaluate distinct aspects of the framework's capabilities, from precise local manipulations to global semantic transformations, while maintaining structural and semantic consistency with the reference image.The non-rigid editing task handles deformable transformations including pose changes and facial expressions. Object addition tests spatial reasoning when introducing new elements. Object removal assesses background reconstruction proficiency. Object replacement evaluates semantic understanding when swapping objects. Attribute modification validates fine-grained control over object characteristics. Background editing examines global scene transformation capacity. These tasks span from localized object manipulations to comprehensive scene transformations, providing thorough evaluation across various editing complexities.

**Quantitative results.** Based on the quantitative results in Table 4, our method demonstrates complementary strengths across editing tasks. The Add operation excels in image preservation (CLIPimg: 0.9740), while Remove achieves the best text alignment (CLIPtxt: 0.2943). Scene editing shows superior directional control (CLIPdir: 0.1195), and Replace maintains balanced performance across metrics. These specialized capabilities provide users with flexible editing strategies tailored to

Table 4: CLIP metrics comparison across different tasks

| Task | CLIPimg | CLIPtxt | CLIPdir |
|------|---------|---------|---------|
| Add | 0.9740 | 0.2869 | 0.0679 |
| Attr | 0.9224 | 0.2857 | 0.0990 |
| Replace | 0.9073 | 0.2885 | 0.1184 |
| Remove | 0.9558 | 0.2943 | 0.0728 |
| Scene | 0.9139 | 0.2811 | 0.1195 |
| Non-rigid | 0.9291 | 0.2837 | 0.0493 |

different requirements—whether prioritizing visual fidelity, semantic accuracy, or edit controllability. The consistent performance validates our method's robustness across diverse editing scenarios.

**Ablation Study** In addition to quantitative metrics, we also qualitatively analyze the ablation results, as shown in Fig. 9. When all blocks are used for injection, the editing signal becomes overly strong, causing the output to almost replicate the reference image and significantly degrading text similarity. Using only single-stream or dual-stream blocks similarly reduces text alignment and introduces inconsistencies with the reference layout. In fact, injecting only at dual-stream blocks even leads to erroneous generation, such as altering the appearance of the crocodile. This occurs because dual-stream blocks lie at lower depths and thus primarily capture low-frequency structure—while preserving global layout, they lack high-frequency information, resulting in distorted object details. Conversely, ablating high-frequency components causes loss of fine details (e.g., the crocodile's teeth), whereas ablating low-frequency components disrupts structural integrity, leading to issues such as inconsistent background regions (e.g., mismatched trees).

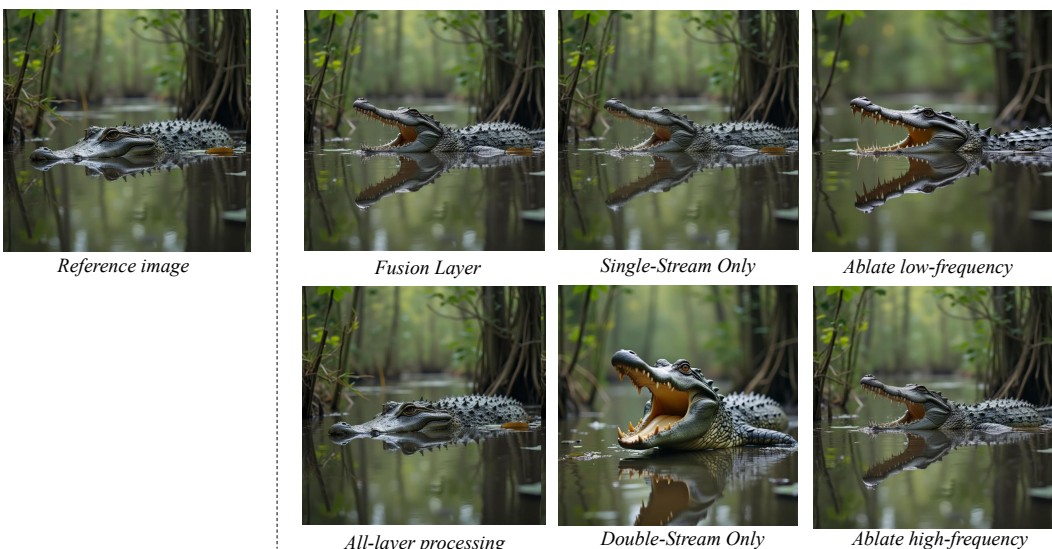

*Reference image* *Fusion Layer* *Single-Stream Only* *Ablate low-frequency*

*All-layer processing* *Double-Stream Only* *Ablate high-frequency*

Figure 9: Visualization results of ablation experiments

**Memory and time usage** Table 5 compares the computational efficiency of different methods, revealing notable performance differences. MagicBrush, built on the U-Net architecture, requires only 16.75 GB of memory with a processing time of 20.38 seconds. In contrast, the other methods are based on the FLUX framework, consuming similar memory (around 35.5–36.1 GB) but with longer processing times ranging from 46 to 50 seconds. Our method achieves superior editing performance while incurring only a 10.8% increase in runtime and a 1.6% increase in memory usage compared to FLUX. While W-Edit delivers competitive editing quality, this efficiency analysis provides practical guidance for selecting methods under given computational resource and time constraints.

Table 5: Memory and time usage

| Method | Memory (GB) | Time (s) |
|---|---|---|
| MaigcBrush | 16.75 | 20.38 |
| FLUX | 35.58 | 46.00 |
| Flow-Edit | 36.09 | 48.00 |
| Stable-Flow | 35.59 | 47.00 |
| W-Edit | 35.59 | 50.00 |

**Qualitative results.** Figures showcase results for object-centric edits. For object addition (Fig. 13), W-Edit exhibits strong spatial reasoning by integrating new objects (e.g., a ceramic vase on a wooden table, hardcover books on a shelf, and lit candles on a birthday cake) into the existing scene geometry without occluding unrelated elements or introducing visual artifacts. The added objects inherit lighting conditions and texture consistency from the reference, ensuring perceptual coherence. In object removal (Fig. 14), the framework not only eliminates target elements (e.g., scattered rocks, a folding chair, park benches) but also inpaints the resulting regions with contextually appropriate content—such as extending grassy terrain or matching wall textures—avoiding the blurred edges or semantic inconsistencies common in baseline methods. For object replacement (Fig. 15), W-Edit preserves fine-grained spatial relationships (e.g., the position of a cake on a plate, bread on a cutting board) while substituting objects with semantically distinct alternatives, demonstrating its ability to disentangle object identity from scene layout. For non-rigid editing (Fig. 16), W-Edit demonstrates remarkable capability in handling complex deformable transformations, including facial expression changes, animal pose adjustments, and body posture modifications, while maintaining structural integrity and avoiding unrealistic distortions. In attribute modification (Fig. 17), the framework achieves precise control over object characteristics such as color, material, and texture, enabling fine-grained edits like altering car colors or modifying book appearances without affecting unrelated regions. Finally, for background editing, W-Edit seamlessly transforms entire environments while preserving consistency of foreground objects, showcasing its capacity for comprehensive scene-level modifications that maintain both semantic coherence and visual plausibility.

As demonstrated in Fig. 21, we comprehensively visualize the editing capabilities of the proposed W-Edit model across the diverse PIE-Bench dataset. Specifically, W-Edit successfully executes various complex operations, including but not limited to: adding or removing objects in a scene, altering the subject's content, pose, background, material, and color, as well as applying overall style transfer. These examples encompass diverse subjects such as animals, humans, and everyday objects, qualitatively validating the model's robustness and precision in understanding and executing complex editing instructions, thereby confirming its powerful and versatile image editing performance.

We also compared W-Edit with the current large-scale pre-trained model FLUX.1 Kontext in Fig 19. As the qualitative comparison in the provided images shows, W-Edit's structural fidelity is not inferior to FLUX.1 Kontext when performing specific edits. For example, in the pose transformations of eagles and dogs, although our method's background consistency is slightly worse than FLUX.1 Kontext, the subject consistency of the edited image is worse than our method. For example, the subjects of the dog and eagle change, and the ability to preserve the subject structure is not as good as our frequency domain method. However, in the cat example, it can be seen that FLUX.1 Kontext's generation is more reasonable and of better quality. But in the bridge example, FLUX.1 Kontext cannot even edit correctly. Therefore, in terms of structural fidelity, W-Edit is not inferior to some current resource-intensive pre-trained editing models, based on the qualitative results.

**The impact of the value of $\eta$ on the results.** To investigate the impact of the hyperparameter $\eta$ on the editing results, we conducted a detailed qualitative analysis, as shown in Fig. 6. In the W-Edit framework, $\eta$ plays a critical role by essentially controlling the amount of frequency sub-band energy retained from the reference image, thereby striking a balance between structural fidelity and editing flexibility. Our analysis indicates that there exists an optimal range for the value of $\eta$. When $\eta$ is set too low (e.g., $\eta$ ¡ 0.4), the model retains too little structural information from the reference image. This leads to issues such as structural collapse or loss of subject identity in the edited results, as illustrated in the first column of Fig. 6. This effect is similar to the "low-frequency removal" scenario in the ablation study in Section 4.3, where the basic composition of the original image cannot be maintained. Conversely, when $\eta$ is set too high (e.g., $\eta$ ¿ 0.8), an excessive amount of frequency information from the reference image is preserved, which overly suppresses the text-guided editing capability. As shown in the last column of Figure X, the editing effect becomes minimal, with the image almost unchanged, resembling the results of the "full-block injection" baseline. As shown in the Tab 3, we found that setting $\eta$ to 0.6 achieves the best balance in most scenarios. Under this configuration, W-Edit is able to firmly preserve the overall structure and subject identity of the source image while effectively responding to textual instructions to produce clear and natural editing effects. The choice of this optimal value is further supported by the quantitative evaluation in Section 4.2, where our method achieved the highest CLIP Score under this setting, demonstrating its superiority in semantic consistency and editing quality. In conclusion, $\eta$=0.6 is established as our default parameter, providing W-Edit with stable and adaptive performance.

**Wavelet Transform Feature Decomposition** To validate the frequency-aware mechanism of the W-Edit framework, we performed Wavelet Transform decomposition and frequency domain analysis on features from various blocks of the FLUX architecture. The $W$ decomposes the feature $F$ into a Low-Frequency Approximation Component ($F_A$) (structure information) and a High-Frequency Detail Component ($F_D$) (texture information). Visualization reveals a non-uniform "Low-Frequency Foundation - High-Frequency Refinement" evolution trend: early blocks are dominated by $F_A$ (establishing structure), while later blocks exhibit significantly increased $F_D$ energy (generating fine-grained texture). Quantitative analysis of the mid-frequency energy further confirms this by showing the feature energy centroid shifting from the low-frequency region to the high-frequency region with increasing layer depth. The strong consistency between the visualization and quantification validates the frequency evolution in the FLUX architecture. Ultimately, this provides a solid theoretical basis for W-Edit's adaptive block selection based on energy metrics, enabling balanced control over structure preservation and detail editing by aligning $FA/FD$ coefficients in the wavelet domain.

**Decomposition Level.** To determine the optimal recursive level for Wavelet decomposition, we systematically visualized the impact of different levels on key features and calculated the resulting energy retention rate, as shown in Fig. 11. Qualitative analysis indicated that the decomposition level significantly affects feature representation. For instance, at Level 1, the low-frequency component effectively captures the image's overall structure, and the high-frequency component retains

clear details. However, as the decomposition level increases (e.g., Level 6), the low-frequency information becomes overly coarse due to repeated downsampling, failing to carry meaningful visual patterns; concurrently, the resolution of the high-frequency components decreases significantly, degrading their ability to represent fine details. Using Parseval's theorem, we performed a quantitative assessment via the energy retention rate, which demonstrated that Level 2 decomposition achieves the optimal balance between structural and detailed information while preserving approximately 85% of the original feature energy. In contrast, Level 1 decomposition might introduce noise due to information redundancy, while excessive levels (such as $\geq 4$) compromise editing quality due to excessive information loss. Based on these qualitative observations and quantitative calculations, we ultimately selected Level 2 decomposition as the core parameter for the W-Edit framework. This choice ensures that the model can fully utilize the low-frequency components to maintain structural consistency and leverage high-quality high-frequency components for detail synthesis in subsequent editing, thus balancing visual fidelity with computational efficiency.

**Layer Comparsion.** Tab. 6 compares the layer selections and performance metrics between Stable-Flow (based on DINOv2 structural saliency) and our frequency energy-based method. The layers [0, 1, 2, 17, 18, 28, 53, 54, 56] were evaluated using two key metrics: Jaccard Similarity Coefficient (0.9) indicates high overlap in layer set composition, demonstrating that our frequency-driven selection aligns closely with Stable-Flow's structural importance criteria. Spearman Rank Correlation Coefficient ( = 0.925) quantifies the agreement in importance rankings derived from ablation studies on edit quality (CLIPimg Score). This strong correlation validates that frequency energy response effectively captures structural significance, supporting our method as a robust alternative to DINOv2-based approaches. The reduction percentages further illustrate comparable performance in stability metrics across methods.

**Block-Wise Frequency Progression.** We utilized a statistically significant subset of 64 unique, high-diversity text prompts (covering concepts like structure, texture, and complex scenes) as input. For each block $l$ and its feature map $F_l$, we calculated the following normalized energy ratios via Fourier Transform. Represents the percentage of total spectral energy contained within the low-frequency radius ($r_{low} = 0.25 \times R_{max}$), directly quantifying the block's anchoring of structural information.Represents the percentage of total spectral energy contained within the high-frequency radius (the outermost ring, ranging from $0.75 \times R_{max}$ to $R_{max}$), directly quantifying the block's refinement of details and texture.

We initially calculated the total energy ($\mathbf{E_{total}}$) of the feature map and obtained $\mathbf{E_{low}}$ and $\mathbf{E_{high}}$ by summing the spectral energy values within their respective frequency bands. We then calculated the $L_2$ norm for all 128 generated samples and averaged the $\mathbf{E_{low}}$ and $\mathbf{E_{high}}$ ratios relative to $\mathbf{E_{total}}$ for each block. Based on this quantitative data, we designed two core visualization charts to illustrate the analysis results intuitively. The Energy Progression 3D Plot uses the Training Step Index and Network Block Index as the horizontal axes and Energy Value as the vertical axis to clearly demonstrate the 3D distribution of $\mathbf{E_{low}}$ and $\mathbf{E_{high}}$ energy across different training phases and network depths. This highlights the shift in energy dominance from low to high frequencies. Furthermore, the Macro Energy Ratio Pie Chart presents the average proportion of low-frequency and high-frequency energy within the total feature representation across the entire diffusion process. These quantitative results strongly confirm a highly significant shift in spectral energy dominance from early to late blocks. We have included a new progression curve figure in the Appendix, shown as Fig 12, that clearly visualizes the evolution trend of the $\mathbf{E_{low}}$ and $\mathbf{E_{high}}$ curves across all blocks. This systematic quantitative and statistical analysis provides the necessary and rigorous theoretical foundation for W-Edit's frequency-aware block-wise injection strategy.

**Decoupling of high and low frequencies.** To further validate this conclusion, we conduct a visualization experiment where only specific frequency bands are injected. As shown in Fig. 18, injecting only low-frequency components alters global structure and layout but fails to refine local details. Injecting only high-frequency components preserves the overall structure while modifying textures and fine-grained attributes.Combining both yields faithful and coherent edits simultaneously maintaining global consistency and enabling precise modifications,This evidence confirms that multi-level frequency control is key to balancing stability and flexibility in editing.

Table 6: Comparison of selected layer.

| Layer ID | $R_sf$ | $R_our$ | $d = R_{sf} - R_{our}$ | $d^2$ | Stable-Flow Drop (%) | Ours Drop(%) |
|----------|--------|---------|------------------------|-------|----------------------|--------------|
| 28 | 6 | 6 | 0 | 0 | 7.2 | 7.5 |
| 54 | 9 | 9 | 0 | 0 | 9.5 | 6.2 |
| 2 | 4 | 5 | 1 | 0 | 8.5 | 8.8 |
| 53 | 7 | 8 | 1 | 1 | 6.8 | 6.8 |
| 0 | 2 | 2 | 1 | 1 | 9.5 | 9 |
| 56 | 8 | 7 | -1 | 1 | 6.8 | 7 |
| 1 | 5 | 3 | 2 | 4 | 7.8 | 9.3 |
| 17 | 1 | 1 | 0 | 0 | 12.1 | 12.5 |
| 18 | 3 | 4 | 1 | 1 | 6.7 | 8.5 |

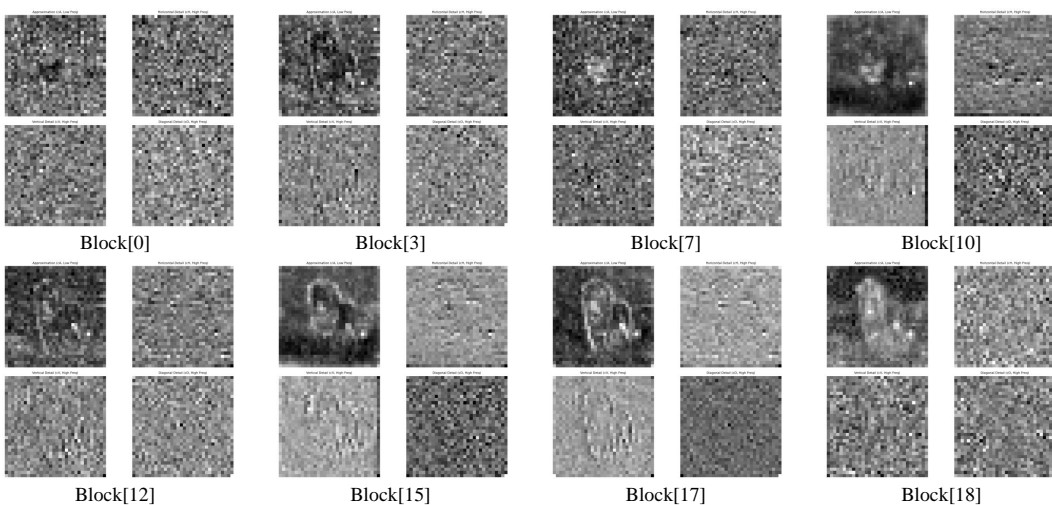

Figure 10: Frequency domain evolution pattern with block.

To verify that the separation of low-frequency structure and high-frequency details in W-Edit enables precise and faithful editing, we conducted the following experiments by injecting low-frequency information and high-frequency information separately.

We designed a rule where, when generating an image based on a text prompt, if we force the preservation of the low-frequency components of the reference image, the generated image is compelled to retain the overall composition and macroscopic appearance of the reference image. Under this strong constraint, the text prompt can only perform subtle semantic replacements on local content, without changing the overall structure or style.

Table 7: Our method yields results for different edit types on the PIE-Bench dataset.

| Categories | Structure$_{Distance}$ | PSNR | LPIPS | MSE | SSIM | CLIPsim | CLIP |
|------------|------------------------|------|-------|-----|------|---------|------|
| random | 0.043 | 20.627 | 0.170 | 0.014 | 0.804 | 24.891 | 31.563 |
| obj | 0.041 | 19.101 | 0.181 | 0.015 | 0.780 | 24.729 | 32.123 |
| add | 0.014 | 24.612 | 0.082 | 0.004 | 0.918 | 24.499 | 31.587 |
| delete | 0.015 | 24.614 | 0.081 | 0.005 | 0.887 | 24.499 | 30.997 |
| content | 0.014 | 25.772 | 0.084 | 0.004 | 0.917 | 24.397 | 31.665 |
| pose | 0.024 | 24.310 | 0.104 | 0.007 | 0.881 | 26.297 | 32.823 |
| change color | 0.020 | 23.956 | 0.096 | 0.007 | 0.903 | 24.888 | 32.659 |
| material | 0.012 | 25.997 | 0.075 | 0.003 | 0.936 | 24.268 | 31.762 |
| background | 0.014 | 25.463 | 0.069 | 0.004 | 0.927 | 23.888 | 31.742 |
| style | 0.013 | 26.167 | 0.085 | 0.011 | 0.913 | 23.955 | 31.480 |

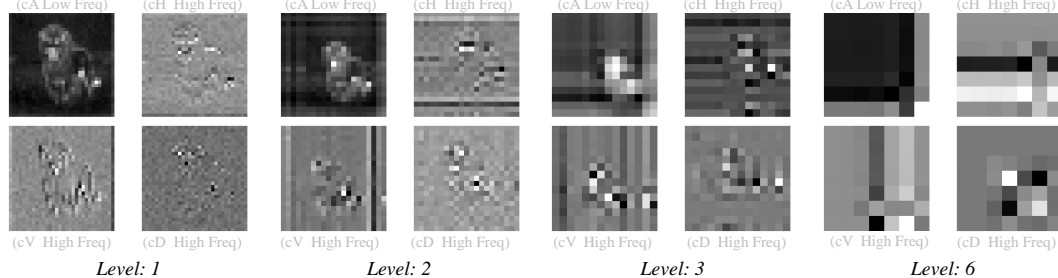

Figure 11: Results of features at different decomposition levels.

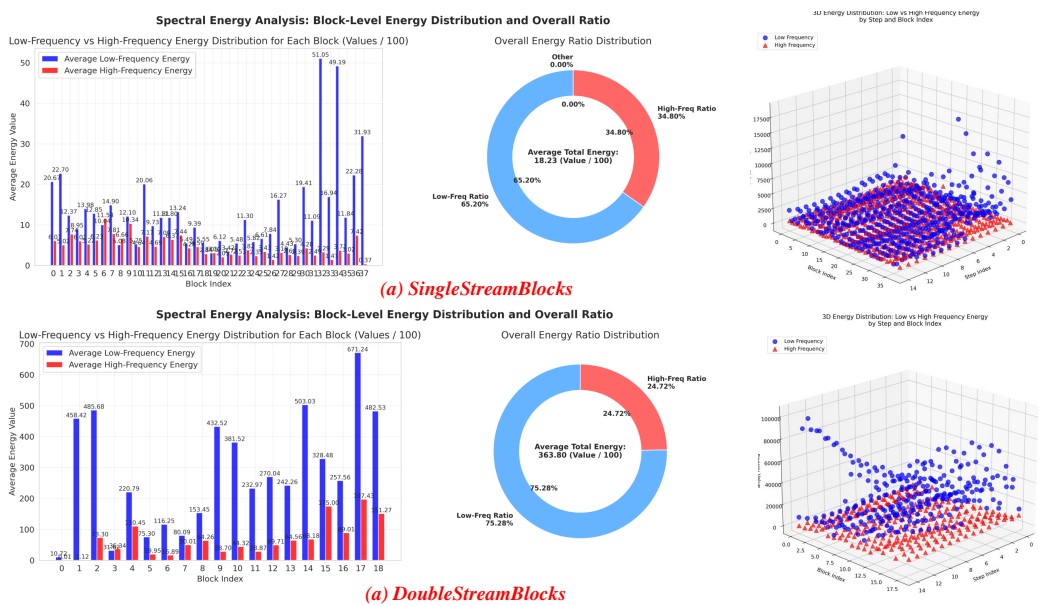

Figure 12: Block-Wise Frequency Progression.

The high-frequency components encode the fine details, textures, outlines, and edges of the objects in the image. By preserving the high-frequency components of the reference image, we are able to precisely lock the geometric shape and contour structure of the object.

Therefore, as shown in Fig. 20: In the upper row, text prompts such as "castle" or "factory" successfully capture local features, like the style of the seaside house, but the image's style, overall background, and layout are still preserved and controlled by the reference image's low-frequency components. In the lower row, when injecting high-frequency information and using a natural landscape (mountain range) as the reference image, the generated image clearly retains the mountain's contour, while its style and texture can be significantly changed according to the text prompt (e.g., "night," "spring," or "desert"). This substantially improves the precision and creativity of local modifications.

Consequently, this verifies that the low-frequency components capture the global structure, macroscopic layout, main color tones, lighting, and artistic style of the image. The high-frequency components encode the fine details, textures, outlines, and edges of the objects in the image. Furthermore, it validates that our method successfully achieves the separation of low-frequency and high-frequency information, significantly enhancing the precision and flexibility of image editing and providing a foundation for achieving excellent editing results.

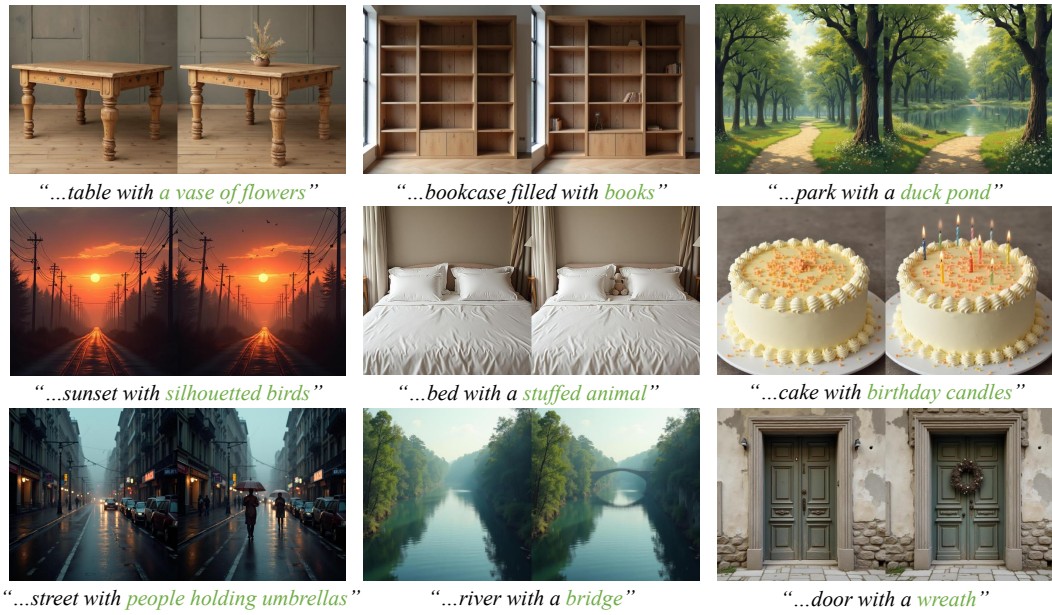

Figure 13: **Object addition results.** W-Edit accurately adds objects specified in the text prompt while preserving the original image structure. For instance, it successfully adds a vase on the table, books on the shelf, and candles to the birthday cake, demonstrating precise spatial understanding and minimal disruption to existing content.

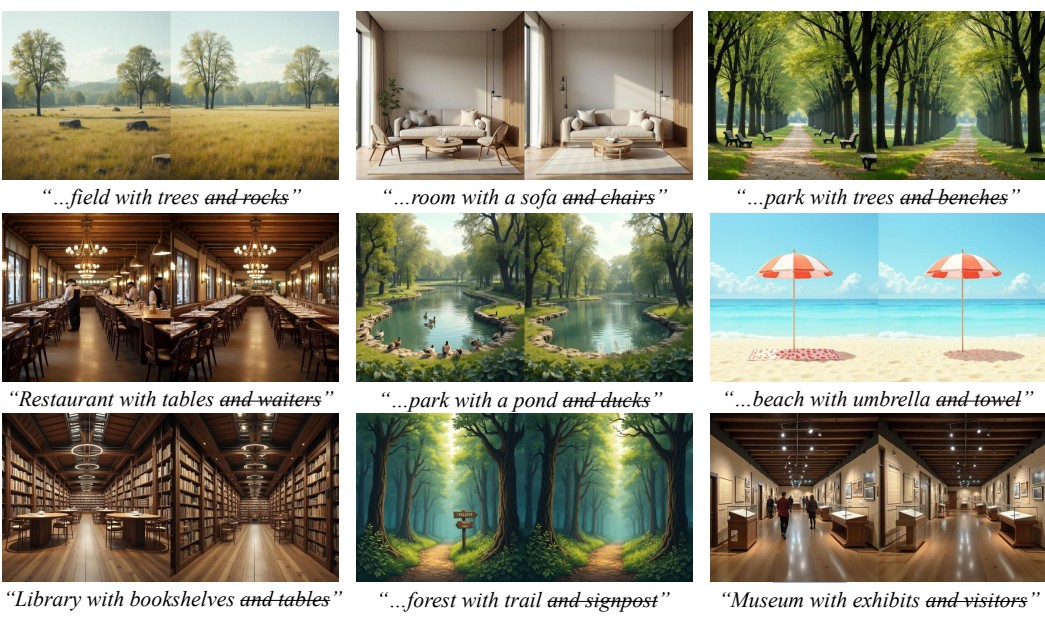

Figure 14: **Object removal results.** W-Edit effectively removes target objects while maintaining scene coherence. Examples include removing rocks from the field, eliminating the chair from the room, and clearing benches from the park. The edited regions show natural inpainting that seamlessly blends with the background.

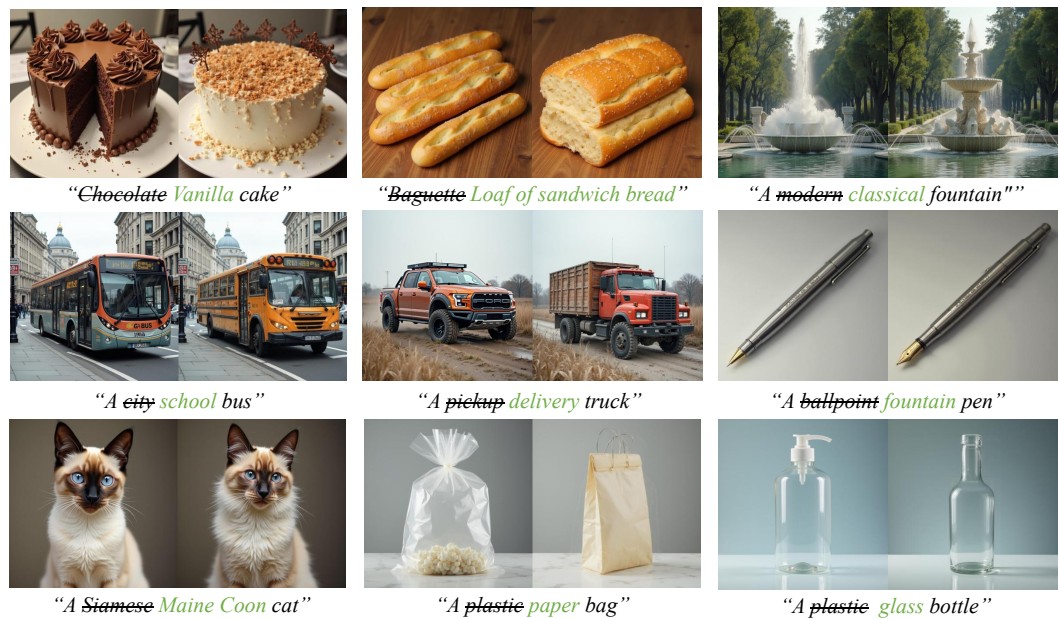

*"Chocolate Vanilla cake"*          *"Baguette Loaf of sandwich bread"*          *"A modern classical fountain""*

*"A city school bus"*          *"A pickup delivery truck"*          *"A ballpoint fountain pen"*

*"A Siamese Maine Coon cat"*          *"A plastic paper bag"*          *"A plastic glass bottle"*

Figure 15: **Object replacement results.** W-Edit demonstrates strong capability in replacing objects with semantically different alternatives while preserving spatial relationships and background consistency. The method successfully replaces chocolate cake with vanilla cake, changes types of bread, and transforms cat appearances, showcasing its flexibility in handling diverse replacement tasks.

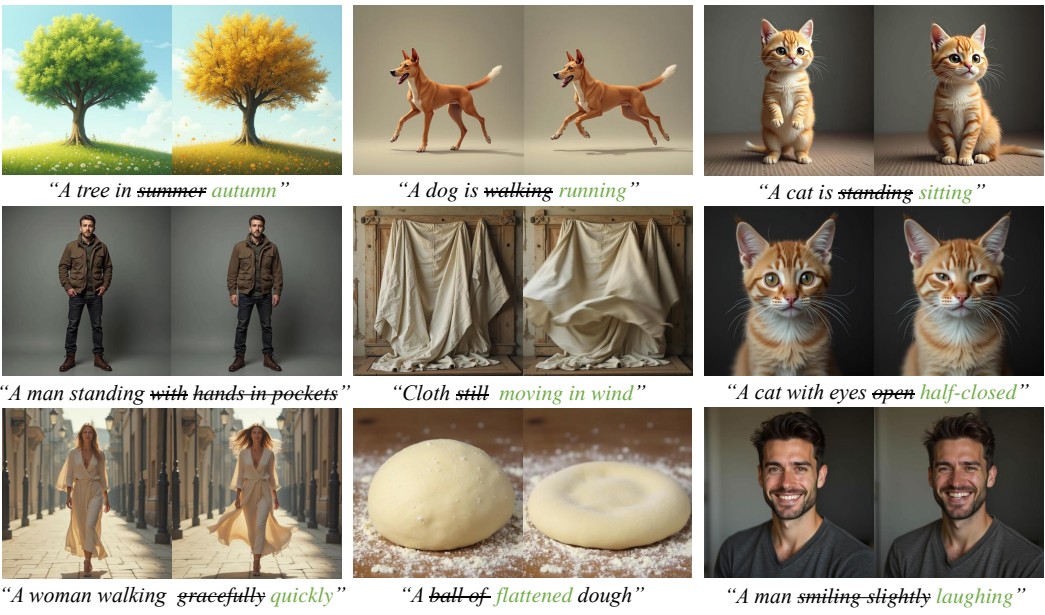

*"A tree in summer autumn"*          *"A dog is walking running"*          *"A cat is standing sitting"*

*"A man standing with hands in pockets"*          *"Cloth still moving in wind"*          *"A cat with eyes open half-closed"*

*"A woman walking gracefully quickly"*          *"A ball of flattened dough"*          *"A man smiling slightly laughing"*

Figure 16: **Non-rigid editing results.** W-Edit demonstrates remarkable capability in handling complex non-rigid transformations while presenting structural integrity. Our method successfully modifies facial expressions, adjusts body postures, and alters animal poses, showcasing its ability to manage deformable objects without introducing artifacts or compromising the original image composition.

## B    REPRODUCIBILITY STATEMENT

Implementation details, evaluation protocols, and dataset descriptions are provided in the main text and appendix. Complete proofs are also included in the main text. The full source code will be released upon acceptance.

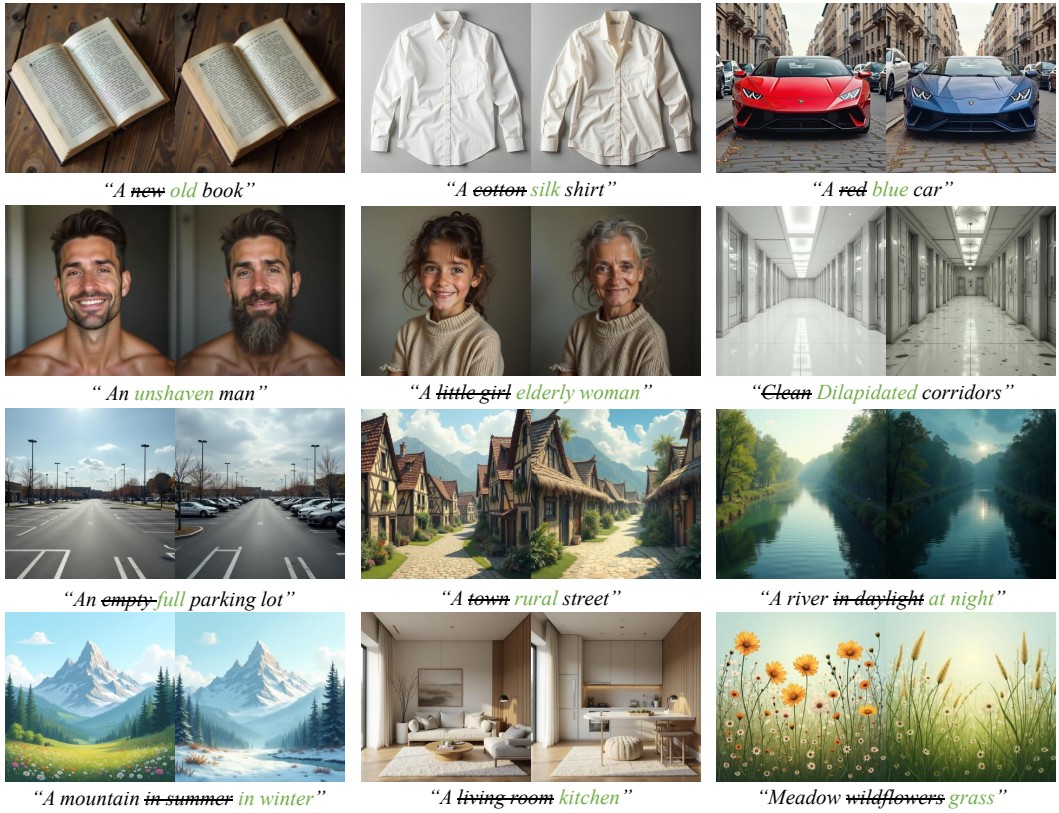

Figure 17: **Attribute modification and scene editing.** W-Edit does well in fine-grained attribute control and comprehensive scene transformation. For attribute modification, it precisely adjusts object properties such as color, material, and texture. For background editing, our method seamlessly replaces entire environments while maintaining consistency of foreground objects.

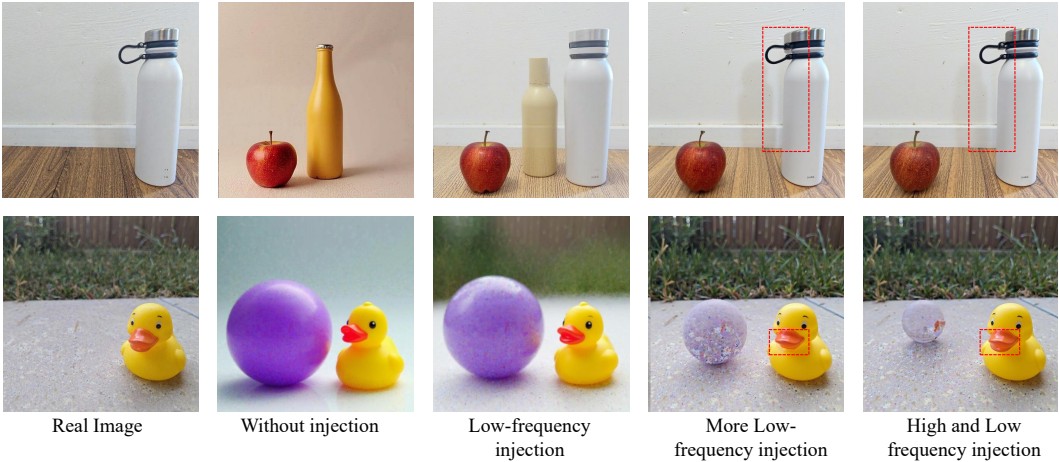

Figure 18: The impact of injecting a specific frequency band on the image.

## C  THE USE OF LLM

Throughout the preparation of this paper, we employed a large language model (LLM) to improve the writing and correct grammatical errors. The LLM was also used as an experimental tool to generate certain prompts for testing purposes.

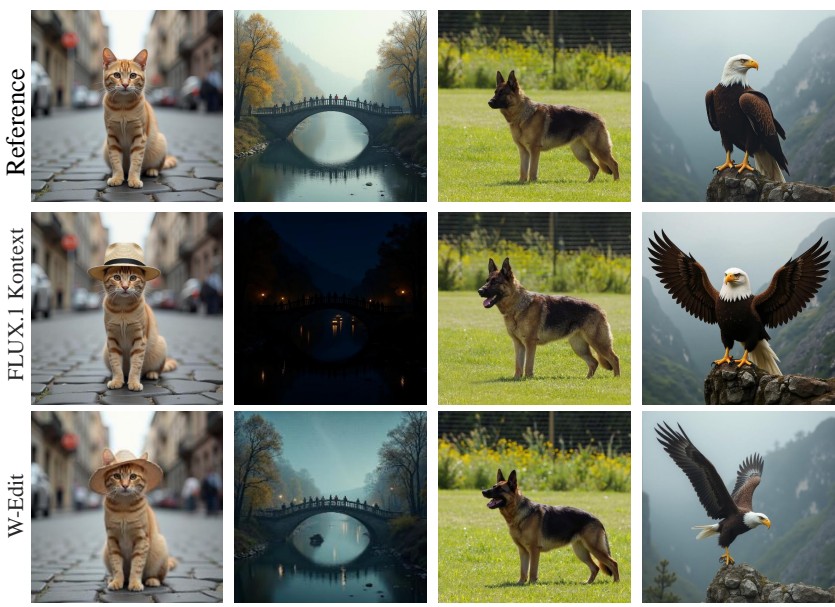

Figure 19: Comparison with FLUX.1 Kontext.

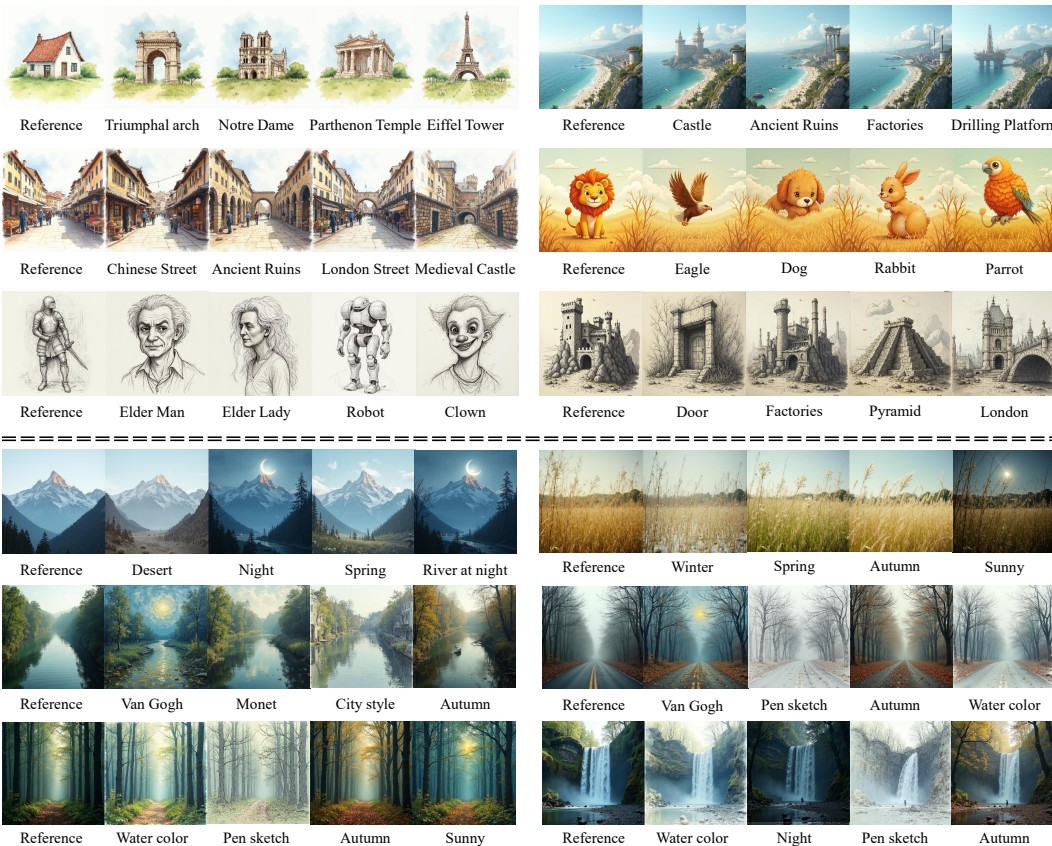

Figure 20: Qualitative Results of Different Frequency Band Replacement Methods. For Low-Frequency Band Replacement, both the appearance and layout of the generated image are controlled by the reference image; for High-Frequency Band Replacement, the reference image controls the contours of the generated image (which is better observed upon magnification).

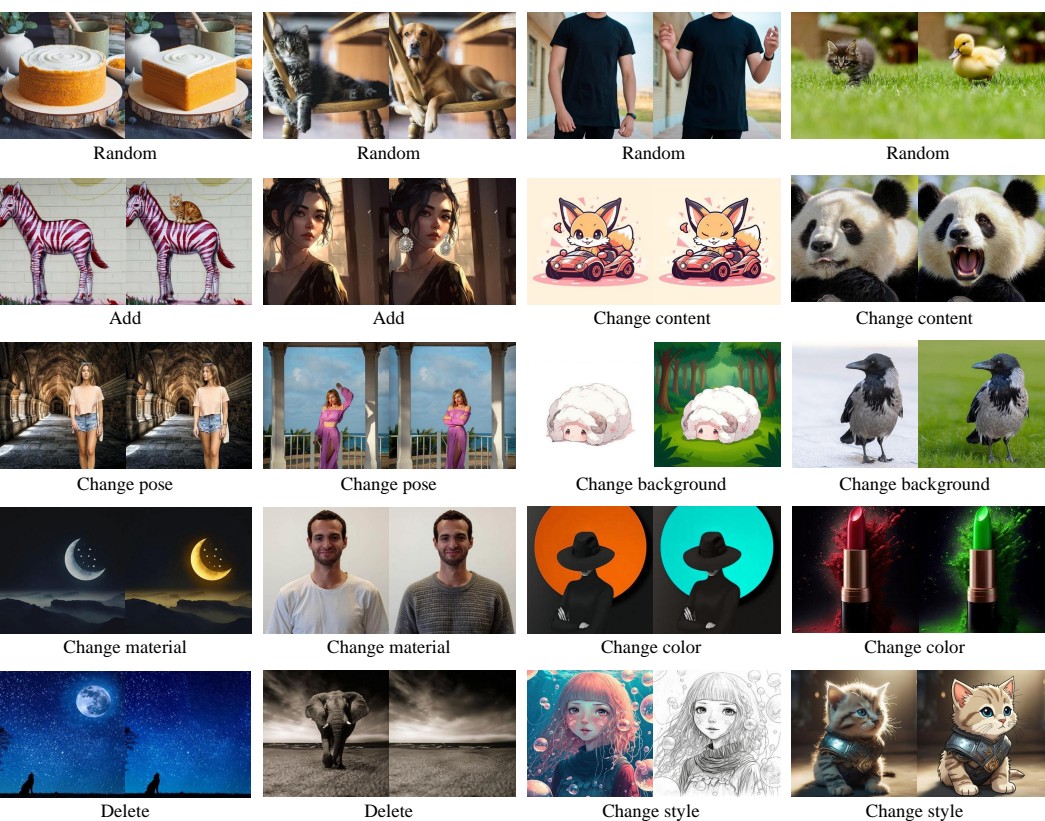

Figure 21: Visualization results of W-Edit for different editing types on PIE-Bench.

