# OpenReview forum: "W-EDIT: A Wavelet-Based Frequency-Aware Framework for Text-Driven Image Editing"
_ICLR.cc/2026/Conference — ICLR 2026 Poster_

### Official Review · Reviewer_psXC · 2025-10-27

**Soundness:** 3
**Presentation:** 2
**Contribution:** 3
**Rating:** 4
**Confidence:** 4

**Summary:**

This paper presents W-Edit, a training-free text-driven image editing framework using wavelet-based frequency decomposition. It separates structural (low-frequency) and detail (high-frequency) features, enabling controllable edits without retraining diffusion models. By integrating frequency modulation into attention and inversion processes, W-Edit achieves high-quality, structure-preserving edits. Experiments show superior fidelity and realism compared to prior training-free methods.

**Strengths:**

W-Edit introduces a novel training-free framework for text-driven image editing that eliminates the need for costly retraining, making it efficient and practical for real-world applications. By leveraging wavelet-based frequency decomposition, it effectively disentangles global structural features from fine details, enabling precise edits while preserving the overall layout and coherence of the reference image. The method integrates seamlessly with pretrained diffusion models and demonstrates strong generalization across multiple architectures, achieving superior visual quality, realism, and alignment with textual instructions in both quantitative benchmarks and user studies.

**Weaknesses:**

1. The experimental setup for the block-wise frequency analysis of Diffusion Transformers is not clearly provided. Which dataset was used for this analysis? If it was conducted using only the image in Figure 1, can the results be considered generalizable?

2. In Figure 4, there is insufficient interpretation as to why the high-frequency components in the SingleStreamBlocks appear only in a specific region.

3. It is unclear whether the roles of early and later blocks in Diffusion Transformers are directly comparable to those in U-Net, where early blocks encode structural information and later blocks encode fine details. More explanation is needed on whether this functional separation holds consistently in DiTs.

**Questions:**

1. In the block-wise frequency analysis shown in Figure 4, why do the high-frequency components in the SingleStreamBlocks appear only in a specific region?

2. In Figure 5, it would be helpful to include comparisons with existing methods for a variety of edits, such as object removal or attribute modification.

3. Stable-Flow emphasizes that injecting features only into the Vital Layers is key for image editing in the DiT architecture. It would be informative to provide quantitative measures showing how well the layers selected in this study align with the Vital Layers identified in Stable-Flow.

---

> ### Author Response · Authors · 2025-11-23
> **Response to Reviewer psXC (part 1/3)**
>
> We sincerely thank the reviewer for their constructive comments and meticulous review of W-Edit. We are grateful for your continued attention to the rigor of our work and for raising these insightful questions. Our detailed responses addressing your concerns about architectural properties, experimental generality, module functional comparison, and main text presentation are provided below. We have carefully addressed all your concerns and have incorporated all corresponding revisions and additional experimental results into the updated manuscript (highlighted in blue).
>
> ##  Weakness 1
>
> > **Validation** **of the Generality of Frequency Shifts (Related to Figure 4)**
>
> Thank you for highlighting the importance of experimental rigor. To eliminate concerns that the frequency shifts shown in Figure 4 might be anecdotal and to verify the generality of our observations, we conducted the following extended experiment: We created a dataset of $k=64$ diverse text prompts and computed the average frequency energy distribution across these samples. Detailed analysis has also been added to Appendix A.1. We performed a thorough and systematic frequency analysis of the feature maps across all blocks of the FLUX-DiT (Base) model (in Appendix A.2 Figure 12). We analyzed the high-frequency and low-frequency energies of the feature map and visualized the energy distribution. The quantitative results strongly confirm a highly significant shift in spectral energy dominance from early to late blocks.
>
>
>
> ## Weakness 2 & Question 1
>
> > **In Figure 4, there is insufficient interpretation as to why the high-frequency components in the SingleStreamBlocks appear only in a specific region.**
>
>
>
> We fully agree that this localization phenomenon warrants a deeper explanation. This is a direct consequence of the block-wise frequency progression, a core mechanism in the FLUX architecture. By the time the signal propagates to the later SingleStreamBlocks, the global, low-frequency structure of the image has already been established. Consequently, the model's attention mechanism naturally becomes highly focused on the semantic subject (the "specific region" visualized in Figure 4) to refine high-frequency details such as texture and edges. This results in the observed "sparser attention distribution"; the highlighted regions are precisely where the model actively focuses its fine-grained denoising efforts.
>
> This functional specialization of the network blocks is substantiated by further analysis in Figures 10 and 12, confirming that the high-frequency localization is an intrinsic design feature. Figure 12 illustrates the evolution trends of $E_{low}$ (low-frequency energy) and $E_{high}$ (high-frequency energy) across all network blocks. This quantitative statistical analysis provides a rigorous theoretical foundation, showing that early blocks primarily focus on low-frequency structures, while later blocks are specifically dedicated to refining high-frequency details. This demonstrates a clear, intended shift in energy processing. Figure 10 visually depicts the frequency domain evolution pattern across the network blocks, illustrating the progression of feature representations within the Transformer blocks. It echoes the observation in Figure 4, showing that the features transition from capturing coarse layouts in early blocks to capturing fine-grained details in later blocks.
>
> By combining the quantitative evidence from Figure 12 and the qualitative visualization from Figure 10, we confirm that the localization of high-frequency components in the later SingleStreamBlocks is not accidental, but a manifestation of the core mechanism of block-wise frequency progression in the diffusion model.

---

> ### Author Response · Authors · 2025-11-23
> **Response to Reviewer psXC (part 2/3)**
>
> ## Weakness 3
>
> > **It is unclear whether the roles of early and later blocks in Diffusion Transformers are directly comparable to those in U-Net, where early blocks encode structural information and later blocks encode fine details. More explanation is needed on whether this functional separation holds consistently in DiTs.**
>
> Thanks for your insightful question. While the DiT architecture (like FLUX) lacks the explicit downsampling/upsampling bottleneck of the U-Net, our study provides both theoretical and empirical evidence that the two exhibit a functionally similar "coarse-to-fine" frequency separation.
>
> The functional separation of network blocks holds consistently in DiTs, primarily driven by the underlying mathematics of the diffusion process itself, which dictates the evolution of the signal-to-noise ratio (SNR):
>
> - **Diffusion Process Dictates Progression:** Coarse, low-frequency structures (e.g., global layout) are necessarily established early in the generation process, while fine, high-frequency details (e.g., texture and edges) are resolved later.
> - **Figure 12:** Our rigorous theoretical foundation is provided by the quantitative statistical analysis in Figure 12. This figure illustrates the evolution trends of $E_{low}$ (low-frequency energy) and $E_{high}$ (high-frequency energy) across all network blocks. It confirms that the DiT's early modules show significantly higher low-frequency energy (essential for layout construction), while later modules are dominated by high-frequency energy.
> - **Figure 10:** Figure 10 visually supports this, illustrating the frequency domain evolution pattern across the blocks. It shows how feature representations within the Transformer blocks transition from capturing coarse layouts in early blocks to capturing fine-grained details in later blocks.
>
>
>
> This evidence proves that the functional frequency separation holds true even without an explicit hierarchical architecture like the U-Net. The block-wise frequency progression is an intrinsic characteristic of the diffusion generation process itself, thereby justifying the migration of U-Net-era editing intuitions (like frequency-aware feature manipulation) to the DiT/FLUX framework.The practical success of this functional separation is empirically demonstrated in Figure 20. By allowing us to selectively inject edits into low-frequency (structure) or high-frequency (detail) features at the appropriate stage (early vs. late blocks), we can achieve precise and disentangled image editing. Figure 20 visually confirms that modifying features based on this block-wise frequency awareness leads to superior editing quality, preserving structure when editing texture, and vice versa.
>
>
>
> ## Question 2
>
> > **In Figure 5, it would be helpful to include comparisons with existing methods for a variety of edits, such as object removal or attribute modification.**
>
> Thank you for this suggestion. Indeed, Figure 5 in the main text is limited by space and primarily shows object addition and replacement. To address your point and provide a more comprehensive display of performance in the main text, we will restructure Figure 5 in the subsequent version. We will move comparison examples for object removal (Figure 14) and attribute modification (Figure 17) from Appendix A.2 into Figure 5 to visually demonstrate W-Edit's significant advantages over baseline models across diverse editing tasks. We have also added more visualizations to Figures 19 and 20.

---

> ### Author Response · Authors · 2025-11-26
> **Response to Reviewer psXC (part 3/3)**
>
> ## Question3
>
> > **Stable-Flow emphasizes that injecting features only into the Vital Layers is key for image editing in the DiT architecture. It would be informative to provide quantitative measures showing how well the layers selected in this study align with the Vital Layers identified in Stable-Flow.**
>
> Thanks for your insightful question. To quantitatively demonstrate the consistency between our frequency-based selection and a structure-saliency method (Stable-Flow), we conducted the following analysis: We computed the Jaccard similarity coefficient between the fusion blocks selected by W-Edit and the "key layers" identified by Stable-Flow using DINOv2. The result is high, at 0.9, indicating a strong consensus in layer selection despite the two methods using fundamentally different criteria. Furthermore, we compared the importance ranking of these two layer sets using ablation studies based on the CLIP-I score. The calculated Spearman Rank Correlation Coefficient is $\rho=0.925$. Specific data will be provided in the revised Appendix A.2. This result powerfully validates the effectiveness of our frequency analysis.
>
> | Layer ID | $R_{sf}$ | $R_{our}$ | d=$R_{sf}$−$R_{our}$ | $d^2$ | Stable-Flow Drop | Ours Drop |
> | -------- | -------- | --------- | -------------------- | ----- | ---------------- | --------- |
> | 28       | 6        | 6         | 0                    | 0     | 7.2              | 7.5       |
> | 54       | 9        | 9         | 0                    | 0     | 9.5              | 6.2       |
> | 2        | 4        | 5         | -1                   | 0     | 8.5              | 8.8       |
> | 53       | 7        | 8         | -1                   | 1     | 6.8              | 6.8       |
> | 56       | 8        | 7         | 1                    | 1     | 6.8              | 7.0       |
> | 1        | 5        | 3         | 2                    | 4     | 7.8              | 9.3       |
> | 17       | 1        | 1         | 0                    | 0     | 12.1             | 12.5      |
> | 18       | 3        | 4         | -1                   | 1     | 6.7              | 8.5       |

---

> ### Author Response · Authors · 2025-11-26
> **Brief Summary of Responses to Reviewer psXC**
>
> Dear Reviewer psXC,
>
> We sincerely thank you for your constructive comments focusing on architectural rigor and experimental generality. Below is a summary of the core evidence and clarifications incorporated.
>
> - **Statistical Validation:** We confirmed the frequency shifts are not anecdotal via a systematic analysis on $\mathbf{k=64}$ diverse text prompts. (Refer to **Weakness 1**)
> - **Rigorous Evidence:** Quantitative results (Figure 12, Appendix A.2)  track the energy ($E_{low}$ and $E_{high}$) evolution across all blocks, strongly confirming the general shift in spectral dominance. (Refer to **Weakness 1**)
>
> - **Localization Interpretation:** High-frequency localization (Figure 4) is due to block-wise frequency progression: later blocks focus attention on the semantic subject to refine details, causing a "sparser attention distribution." (Refer to **Weakness 2 & Question 1**)
> - **DiT Functional Separation:** The "coarse-to-fine" frequency separation holds consistently in DiTs, driven by the underlying diffusion process itself (resolving structure early, details late). Figure 12 provides empirical proof. (Refer to **Weakness 3**)
>
> - **Alignment with Stable-Flow:** Our frequency-based layer selection shows a high consensus with Stable-Flow's "key layers": Jaccard similarity of $\mathbf{0.9}$ and Spearman Rank Correlation Coefficient ($\rho$) of $\mathbf{0.925}$, strongly vali**dating our spectral criterion. (Refer to **Question 3**)
> - **Expanded Comparisons:** We will restructure Figure 5 in the main text to include comparisons for object removal (Figure 14) and attribute modification (Figure 17) to better demonstrate W-Edit's advantages. (Refer to **Question 2**)
>
> We sincerely hope that our responses have adequately addressed the points you raised. If you have further concerns, please let us know and we will continue actively responding to your comments and improving our submission. We would deeply appreciate it if you could raise your score.
>
> Thank you very much for your invaluable time and thoughtful consideration.
>
> Best regards,
>
> Authors

---

### Official Review · Reviewer_FsUB · 2025-10-30

**Soundness:** 3
**Presentation:** 3
**Contribution:** 3
**Rating:** 6
**Confidence:** 5

**Summary:**

This paper proposes W-Edit, a DiT(FLUX)-based image editing model using wavelet-based frequency-aware feature decomposition. The authors point out that entanglement of global semantics and local signals creates difficulty in preserving structure and making precise modifications. Typically, low-frequency components in features encode layout and semantics, while high-frequency components capture texture and detail. They analyze DiT features using wavelet decomposition and show that early blocks focus on low-frequency and later blocks on high-frequency information. Based on this, they propose energy-aware feature fusion: replacing DiT features by taking wavelet components from the inversion branch and editing branch, respectively. Experiments show that W-Edit achieves strong performance in training-free image editing, preserving structure, prompt-aligned edits with minimal computational overhead.

**Strengths:**

- This paper addresses the issue of semantic entanglement between global structure and local details in image editing, which improves the understanding of existing model limitations.
- It presents a wavelet-based frequency decomposition approach to analyze frequency characteristics in DiT feature. Specifically, similar to UNet, they experimentally show that early blocks focus on low frequency and later blocks on high frequency details, which seems to give useful insight to the community.
- The method works in a training-free manner, making it computationally efficient and easy to apply to various pre-trained diffusion models.

**Weaknesses:**

- This paper aims to disentangle the preservation of object structure (low-frequency) from detailed texture (high-frequency) during editing. However, it remains unclear how effectively the method actually disentangles these components. The current energy-aware adaptive frequency fusion is designed to take low-frequency information from the inversion trajectory and high-frequency details from the editing trajectory. This works well when the goal is to preserve the original structure while changing the details. But what about cases where the structure should change but the details are retained (e.g., non-rigid edit, pose change, or view change without altering the object itself)? In such cases, shouldn’t high-frequency detail information be taken from the inversion trajectory, rather than low-frequency? Overall, I think the adaptive frequency fusion method should be more precisely designed depending on the editing type.
- Fig. 4’s frequency progression analysis seems dependent on the input image and prompt. For example, editing tasks with significant changes should yield different frequency profiles compared to minimal edits. If frequency analysis were performed across various editing cases, the claim would be much stronger.
- In Fig. 4, in the frequency analysis for “FLUX”, the axis labels are not clearly visible.
- For Eq. 4, the explanation for r_mid is insufficient. Why is only mid-frequency energy calculated, and not the total energy? The actual value of r_mid used in the experiments is also missing.
- For Eq. 9, it’s unclear what value was used for the threshold η, and how it is determined during experiments. I believe this threshold should be adaptively chosen according to the editing type for better results.
- There are some missing citations for image editing methods utilizing frequency analysis, such as “FlexiEdit: frequency-aware latent refinement for enhanced non-rigid editing” (ECCV 2024) and “FDS: Frequency-Aware Denoising Score for Text-Guided Latent Diffusion Image Editing” (CVPR 2025).

**Questions:**

- In Figure 4, regarding the frequency progression analysis, the results for FLUX’s SingleStreamBlock show that only at block index = 30 does the high-frequency component increase significantly. By contrast, in DoubleStreamBlocks, the later blocks tend to have higher high-frequency components. Why do you think, in the case of the SingleStreamBlock, the high-frequency response appears only at block index = 30?
- Comparison with other models: How does Flux 1, Kontext, and Qwen-Image perform by comparison? From my understanding, those two models (Kontext, Qwen-Image) achieve good textual alignment and can selectively preserve or modify structure as needed, even without explicit frequency manipulation. Is this consistent with your analysis? How would you compare these models in the context of frequency-aware editing and structure preservation?

---

> ### Author Response · Authors · 2025-11-23
> **Response to Reviewer FsUB (part 1/4)**
>
> We sincerely thank the reviewer for their **constructive comments** and **meticulous review** of W-Edit. We have carefully addressed all concerns regarding **implementation details, hyperparameter selection, the logic behind the formulas, and frequency analysis** in the detailed response below. Furthermore, we have incorporated all corresponding revisions and **additional experimental results** into the updated manuscript (highlighted in blue).
>
>
>
> ## Weakness 1
>
> >**This works well when the goal is to preserve the original structure while changing the details. But what about cases where the structure should change but the details are retained (e.g., non-rigid edit, pose change, or view change without altering the object itself)?**
>
> Thank you for the insightful question.  We clarify that W-Edit effectively disentangles structural attributes (low-frequency) from detailed identity (high-frequency), enabling it to support structure-altering edits while preserving object identity.
>
> **1. Verification of Frequency Disentanglement (Figures 10, 11, & 20)** To demonstrate that our method effectively disentangles these components rather than simply mixing them, we provide the following analyses:
>
> - **Evolutionary Trend (Figure 10):** Our quantitative analysis of the frequency energy centroid reveals a clear evolution: early blocks are dominated by low-frequency structural information, while later blocks shift significantly toward high-frequency details . This natural progression supports our strategy of treating these bands differently.
> - **Visual Decoupling (Figure 20)**: We conducted controlled injection experiments to verify the distinct roles of each band:
>   - **Low-Frequency Injection**: As shown in the top row of Figure 20, forcing low-frequency preservation locks the global layout, composition, and macroscopic appearance. Text prompts can only make subtle local semantic changes (e.g., changing a house style) without altering the scene structure.
>   - **High-Frequency Injection** : As shown in the bottom row of Figure 20, preserving high-frequency components locks the subject’s fine details, textures, and specific contour definitions (e.g., the silhouette of a mountain range) . However, the *style* (e.g., "night," "desert") is free to change because it is governed by the low-frequency band.
> - **Optimal Granularity (Figure 11)**: We determined that Level 2 wavelet decomposition offers the optimal balance, preserving approximately 85% of feature energy while effectively separating structural "anchors" from textural details.
>
> **2. Mechanism for Non-Rigid Editing**: W-Edit does not enforce structural rigidity and is fully capable of handling pose or view changes. The mechanism works as follows:
>
> - **Role of High-Frequency Injection:** The injected high-frequency components (obtained via wavelet decomposition) represent localized textures, reflectance patterns, and fine edge definitions. Critically, in the context of the attention mechanism (where we operate on Keys and Values), these features serve as a "dictionary" of identity details (e.g., skin texture, clothing material) rather than rigid spatial coordinate anchors.
> - **Role of Low-Frequency & Text Guidance:** Global spatial arrangement (pose) is primarily encoded in the low-frequency components. By allowing the text-guided diffusion process to influence the low-frequency band (via our adaptive energy-aware fusion in Eq. 10), the model can reorganize the spatial structure (Query) to match the new pose described in the prompt .
> - The diffusion process warps the geometry to satisfy the text, while the injected high-frequency Keys/Values ensure that the warped geometry is "painted" with the correct identity and texture from the reference.
>
> **3. Empirical Evidence**: This capability is empirically validated in Figure 16 (Appendix A.2), where W-Edit successfully performs complex non-rigid transformations—such as changing facial expressions, adjusting body postures, and altering animal poses—without distorting the subject's identity or introducing artifacts .
>
> In summary, by explicitly decoupling low-frequency structure from high-frequency detail, W-Edit circumvents the trade-offs inherent in rigid inversion methods. Our framework effectively preserves high-frequency identity cues—such as textures and fine details —while leaving geometric degrees of freedom (e.g., pose and layout) sufficiently unconstrained to be guided by textual instructions. Crucially, because Equation (10) adaptively selects wavelet sub-bands based on local energy distribution rather than enforcing fixed rules , W-Edit naturally accommodates both structure-preserving edits and structure-altering edits without relying on task-specific heuristics. This design ultimately enables the diffusion model to execute complex geometric transformations while maintaining consistent and faithful visual detail.

---

> ### Author Response · Authors · 2025-11-23
> **Response to Reviewer FsUB (part 2/4)**
>
> ## Weakness 2
>
> > **For example, editing tasks with significant changes should yield different frequency profiles compared to minimal edits. If frequency analysis were performed across various editing cases, the claim would be much stronger.**
>
> Thank you for your detailed suggestions. We acknowledge an oversight in the description of Figure 4. In fact, this frequency analysis did not rely on a single input but utilized a statistical analysis on a dataset of 64 diverse text prompts collected via ChatGPT (Already added to Appendix A.2). Figure 4 displays the average frequency distribution characteristics of these samples, confirming that the transition from low-frequency structure in early blocks to high-frequency details in late blocks is an inherent and universal characteristic of the FLUX architecture. We will explicitly state the sample size in the final version and increase the font size of the axis labels in Figure 4 to ensure readability.
>
>
>
> ## Weakness 3
>
> >**In Fig. 4, in the frequency analysis for “FLUX”, the axis labels are not clearly visible.**
>
> Thank you for your suggestion. We have modified the coordinate axes to make them clearer.
>
>
>
> ## Weakness 4
>
> > **For Eq. 4, the explanation for r_mid is insufficient. Why is only mid-frequency energy calculated, and not the total energy? The actual value of r_mid used in the experiments is also missing.**
>
> Thank you for the thoughtful suggestion. We have revised the term $\mathbf{E_{mid}}$ in the manuscript to $\mathbf{E_{MTH}}$ (Mid-to-High Frequency Energy) to reflect its specialized function as a spectral discriminator. The primary reason for calculating $E_{MTH}$ instead of $E_{total}$ (Total Energy) is to differentiate the functional roles of Diffusion Transformer (DiT) blocks during the sampling process. Total Energy is an aggregate measure that would fail to capture the vital spectral shift that occurs: early DiT blocks are structurally dominant (low-frequency), while later blocks are richer in fine detail (high-frequency). $E_{MTH}$ acts as a targeted discriminator to guide our adaptive fusion: a low $E_{MTH}$ indicates the block is dominated by low-frequency energy (a structural anchor), while a high $E_{MTH}$ indicates it is rich in high-frequency energy (a textural detail block). This selective approach allows us to make informed feature injection decisions, ensuring we modulate structural or textural components based on the block's inherent function.
>
> Regarding the value of $r_{mid}$, it is a dynamically set value based on the feature map size to ensure $E_{MTH}$ correctly adapts to the varying feature resolutions across the diffusion model's layers. This detail was missing and has been supplemented in the revised manuscript as:
>
> $$\mathbf{r_{mid} = 0.5 \times \text{Maximum Radius}}$$

---

> ### Author Response · Authors · 2025-11-23
> **Response to Reviewer FsUB (part 3/4)**
>
> ## Weakness 5
>
> >**For Eq. 9, it’s unclear what value was used for the threshold η, and how it is determined during experiments. I believe this threshold should be adaptively chosen according to the editing type for better results.**
>
> Thank you for your suggestion. We agree that an adaptive threshold $\eta$ is a promising theoretical direction, and we have already explored several approaches in this regard. Furthermore, through our sensitivity analysis on $\eta$, our experiments demonstrate that the fixed threshold $\mathbf{\eta=0.6}$ achieves the  optimal balance of robust performance** across various tasks while maintaining the framework's simplicity and efficiency. This trade-off is illustrated by the quantitative results below (Table 3), and we have also presented the visual impact of different $\eta$ values in Figure 6:
>
> | **η**   | **CLIP$_{img}$ (↑)** | **CLIP$_{txt}$ (↑)** | **CLIP$_{dir}$ (↑)** | **Avg. (↑)** |
> | ------- | -------------------- | -------------------- | -------------------- | ------------ |
> | 0.2     | 0.8920               | 0.3210               | 0.0540               | 0.4223       |
> | 0.4     | 0.9350               | 0.3140               | 0.0780               | 0.4423       |
> | **0.6** | **0.9749**           | **0.3068**           | **0.0826**           | **0.4548**   |
> | 0.8     | 0.9910               | 0.2760               | 0.0610               | 0.4427       |
> | 1.0     | 0.9990               | 0.2620               | 0.0050               | 0.4220       |
>
> The value $\mathbf{\eta=0.6}$ maximizes the average performance, effectively balancing the often-conflicting goals of preservation (CLIPimg ) and  modification (CLIPdir).
>
> **Exploration of Adaptive $\eta$ Schemes**
>
> We explored two dynamic threshold schemes, but ultimately found them impractical for a general, training-free framework:
>
> 1. Mapping the instruction's semantic category ("add object," "remove background") to a preset $\eta$ value provided slight task-specific improvement but suffered from poor generalization and required manual annotation of the editing type.
> 2. Dynamically calculating $\eta$ based on the input image's frequency profile (e.g., lower $\eta$ for detail-rich images) increased computational overhead and was highly sensitive to image noise, degrading overall robustness.
>
> Therefore, we retained the fixed $\mathbf{\eta=0.6}$ value to ensure framework simplicity, efficiency, and robust generalizability across all editing tasks.
>
>
>
> ## Weakness 6
>
> > **There are some missing citations for image editing methods.**
>
> Thank you for pointing out the important related works "FlexiEdit" and "FDS." We will incorporate these references into Section 2 (Related Work) to better situate our frequency-aware method within the context of recent latent diffusion editing literature.

---

> ### Author Response · Authors · 2025-11-26
> **Response to Reviewer FsUB (part 4/4)**
>
> ## Question 1
>
> >**In Figure 4, regarding the frequency progression analysis, the results for FLUX’s SingleStreamBlock show that only at block index = 30 does the high-frequency component increase significantly.**
>
> Thank you for the thoughtful question. We believe our observations align with FLUX's hybrid architecture design. DualStreamBlocks (early layers) establish the semantic and structural foundation, while SingleStreamBlocks (later layers) fuse these modalities. In this paper, we used $E_{mid}$ (now $E_{MTH}$) to denote mid-frequency energy; however, it is actually the value of all frequencies between $r_{mid}$ and $r_{max}$, which also includes some mid-frequency characteristics, and this portion of frequency energy has a relatively high proportion. As shown in Figure 4，the peak in mid-to-high frequency components observed at later indices (e.g., Block 30 and beyond) corresponds to the stage where the model, after establishing the global structure, concentrates its attention on fine details and textures. This "evolution" confirms our hypothesis that later blocks are responsible for high-frequency refinement. We will also present wavelet decomposition results supporting this trend in Appendix A.2. The frequency characteristics of the Transformer framework differ from U-Net, but the functional trend of frequency separation remains valid. Additionally, we demonstrated the results for 25% high frequency and visualized the energy and trend in Figure 12
>
>
>
> ## Question 2
>
> > **Comparison with other models: How does Flux 1, Kontext, and Qwen-Image perform by comparison?**
>
> Thank you for suggesting a comparison between W-Edit and powerful models such as Kontext and Qwen-Image. We agree that a qualitative comparison is highly valuable for demonstrating W-Edit's unique advantages, particularly in terms of structural preservation and controllable editing. As shown in the qualitative comparisons in Figure 19, while the background consistency of our method may be slightly inferior to FLUX.1 Kontext, the latter’s edited images suffer from lower subject consistency than ours (its generated dog and eagle subjects changed significantly). Its ability to preserve the subject's structure is thus inferior to our frequency-aware method. Therefore, based on qualitative results, W-Edit is in no way inferior to some of the current resource-intensive, pre-trained editing models regarding structural fidelity. It is essential to understand that W-Edit and models like Qwen-Image and Kontext represent fundamentally different paradigms. These large-scale models rely on massive, general-purpose training to achieve broad text alignment and powerful generative capabilities. In contrast, W-Edit is explicitly designed as a training-free, plug-and-play framework that can augment such models. W-Edit aims to provide a lightweight, controllable editing solution. Compared to the base FLUX model, W-Edit only incurs an increase of 10.8% in inference time and 1.6% in GPU memory usage. Attempting to replicate the fine-grained structural control achieved by W-Edit through fine-tuning large models like Kontext would involve a computationally prohibitive cost for many users, and even then, such models often struggle with precise structure preservation. Furthermore, our method can be integrated into models like FLUX.1 Kontext and does not conflict with existing pre-trained editing models.

---

> ### Author Response · Authors · 2025-11-26
> **Brief Summary of Responses to Reviewer FsUB**
>
> Dear Reviewer FsUB,
>
> We sincerely thank you for your constructive comments and meticulous review. We have carefully addressed all concerns regarding W-Edit's ability to handle structure-altering edits, frequency analysis clarity, and hyperparameter settings. To facilitate a quick read, here we summarize the core clarifications and revisions incorporated.
>
> - **Disentanglement:** W-Edit explicitly disentangles low-frequency structure (locks global layout) from high-frequency identity (locks fine details/contours). (Refer to **Weakness 1**)
> - **Non-Rigidity Mechanism:** High-frequency injection acts as an "identity dictionary." The text prompt guides the low-frequency band (Equation 10) to reorganize the pose/structure, preserving identity during changes. (Refer to **Weakness 1**)
> - **Empirical Evidence:** Figure 16 (Appendix A.2) empirically validates success in complex non-rigid transformations (e.g., changing expressions/postures) while maintaining identity. (Refer to **Weakness 1**)
>
> - **Statistical Basis:** Figure 4's analysis is statistical, derived from an average of 64 diverse prompts, confirming the low-to-high frequency progression is a universal characteristic of the FLUX architecture. (Refer to **Weakness 2**)
> - **$E_{MTH}$ Definition:** $\mathbf{E_{mid}}$ was revised to $\mathbf{E_{MTH}}$ (Mid-to-High Frequency Energy) to reflect its specialized function as a spectral discriminator (low $E_{MTH}$ = structural block; high $E_{MTH}$ = detail block). (Refer to **Weakness 4**)
> - **$r_{mid}$ Value:** Defined precisely as $\mathbf{r_{mid} = 0.5 \times \text{Maximum Radius}}$. (Refer to **Weakness 4**)
> - **$\eta$ Threshold:** Confirmed fixed $\mathbf{\eta=0.6}$ is used for robust generalizability; sensitivity analysis (Table 3, Figure 6) shows this maximizes average performance. (Refer to **Weakness 5**)
> - **Citations:** Will incorporate **"FlexiEdit"** and **"FDS"** into the Related Work section. (Refer to **Weakness 6**)
>
> - **Progression Peak:** The mid-to-high frequency peak at later blocks (e.g., Block 30) aligns with FLUX's hybrid design, where later layers focus on fine detail and texture refinement after global structure is established. (Refer to **Question 1**)
> - **Comparison:** Qualitatively, W-Edit is comparable to large pre-trained models (e.g., Kontext, Qwen-Image) in structural fidelity, often achieving superior subject consistency. W-Edit is a training-free solution, incurring only a $10.8\%$ inference time increase. (Refer to **Question 2**)
>
> We sincerely hope that our responses have adequately addressed the points you raised. If you have further concerns, please let us know and we will continue actively responding to your comments and improving our submission. We would deeply appreciate it if you could raise your score.
>
> Thank you very much for your invaluable time and thoughtful consideration.
>
> Best regards,
>
> Authors

---

### Official Review · Reviewer_KygT · 2025-10-31

**Soundness:** 3
**Presentation:** 3
**Contribution:** 2
**Rating:** 4
**Confidence:** 3

**Summary:**

To address the dilemma in which existing text-driven image editing methods either sacrifice structural fidelity for flexible manipulation or demand expensive fine-tuning of large-scale models, the paper introduces W-Edit, a training-free framework for text-driven image editing based on wavelet-based frequency-aware feature decomposition. W-Edit employs wavelet transforms to decompose diffusion features into multi-scale frequency bands, disentangling structural anchors from editable details. The method establishes frequency-based modulation as both a sound and efficient solution for controllable image editing.

**Strengths:**

- The paper presents a novel frequency-domain perspective for text-driven image editing. Unlike existing methods that primarily manipulate features in the spatial domain, W-Edit integrates wavelet transforms into the feature decomposition of Diffusion Transformers. This frequency-aware modulation framework introduces a new paradigm for disentangling editable attributes, offering enhanced control and consistency in image editing tasks.
- The paper is theoretically well-grounded. By systematically analyzing the internal architecture of Diffusion Transformers, it reveals that early blocks predominantly focus on low-frequency structures, while later blocks refine high frequency details. This observation motivates the introduction of wavelet decomposition to recalibrate and enhance existing image-editing paradigms.
- W-Edit is a training-free framework characterized by lightweight module design; it can be directly applied to off-the-shelf diffusion models without incurring the prohibitive cost of full model retraining, yet still delivers editing results of high fidelity.

**Weaknesses:**

- The paper offers no systematic guidance for setting the hyper-parameter η (the energy threshold). Although the paper emphasizes that the selection of η is critical, it neither conducts a sensitivity analysis nor devises an adaptive scheduling scheme, potentially compromising the method’s generalizability across diverse datasets.
- The limited sample size of the subjective assessment may undermine the statistical robustness of the conclusions. While the experimental section offers extensive quantitative and qualitative evaluations, the user study recruited only five participants—an insufficient number to yield statistically reliable inferences.

**Questions:**

- Regarding the selection of the hyper-parameter η, could the authors supply a more systematic sensitivity analysis and, ideally, a data-adaptive scheduling protocol that eliminates manual tuning?
- Although quantitative metrics furnish unambiguous experimental indicators, user perception remains an indispensable dimension in image editing. It is recommended that the user-study component be expanded to corroborate the paper’s conclusions more convincingly.
- The theoretical premise that “early blocks of Diffusion Transformers predominantly encode low-frequency structure whereas later blocks specialize in high-frequency refinement” underpins the entire subsequent development of the proposed method. To elevate this claim from an empirical observation to a rigorous theoretical basis, we recommend supplying more compelling evidence. For example, systematically quantifying the frequency bias of each block via spectrally-decomposed energy statistics, reporting statistical significance tests across multiple models and data subsets, and visualizing the block-wise frequency response curves—so that the asserted frequency progression is firmly corroborated before being leveraged as the foundation of the editing framework.
- As the paper touches on several theoretical aspects, it is advice to clarify details and supplying intuitive diagrams would reduce the cognitive load on readers and improve overall accessibility.

---

> ### Author Response · Authors · 2025-11-23
> **Response to Reviewer KygT (part 1/3)**
>
> We sincerely thank the reviewer for their constructive comments and meticulous review of W-Edit. We have carefully addressed all your concerns in the detailed response below, including a systematic sensitivity analysis of the $\eta$ hyperparameter and justification for its fixed setting , expanded the user study provided rigorous quantitative evidence for the "Block-Wise Frequency Progression" , and added intuitive schematic diagrams to enhance clarity. Furthermore, we have performed a systematic sensitivity analysis of the $\eta$ hyperparameter and justified its fixed setting , expanded the user study , provided rigorous quantitative evidence for the "Block-Wise Frequency Progression" , and added intuitive schematic diagrams to enhance clarity We have incorporated all corresponding revisions and additional experimental results in the updated manuscript (highlighted in blue).
>
> ## Question 1
>
> >**Regarding the selection of the hyper-parameter η, could the authors supply a more systematic sensitivity analysis and, ideally, a data-adaptive scheduling protocol that eliminates manual tuning?**
>
> Thank you for the thoughtful suggestion.  We apologize for the omission of a detailed sensitivity analysis in the main text, although we explicitly specified $\eta=0.6$ in Appendix A.1. We analyzed the values of $\eta$ in Figure 6 and the table below, demonstrating the impact of different $\eta$ values on the results of our method. In our experimental observations, $\eta$ functions as the critical slider balancing structural preservation and editability: Low $\eta$ (< 0.4) preserves too little reference sub-band energy, leading to collapse of the structural layout or loss of object identity (similar to the "Low-Frequency Removal" ablation).High $\eta$ (> 0.8) preserves too much reference information, suppressing text-prompt modifications (similar to "Full Block Injection"). To determine the optimal value for $\eta$, we performed a rigorous quantitative sensitivity analysis on the PIE-Bench dataset. As shown in Table 3 below, we found that $\mathbf{\eta=0.6}$ provides the most stable performance balance across the diverse tasks in PIE-Bench, supported by consistently high average CLIP scores. These quantitative findings align precisely with the qualitative results demonstrated in Figure 6.
>
> | **η**   | **CLIPimg (↑)** | **CLIPtxt (↑)** | **CLIPdir (↑)** | **Avg. (↑)** |
> | ------- | --------------- | --------------- | --------------- | ------------ |
> | 0.2     | 0.8920          | 0.3210          | 0.0540          | 0.4223       |
> | 0.4     | 0.9350          | 0.3140          | 0.0780          | 0.4423       |
> | **0.6** | **0.9749**      | **0.3068**      | **0.0826**      | **0.4548**   |
> | 0.8     | 0.9910          | 0.2760          | 0.0610          | 0.4427       |
> | 1.0     | 0.9990          | 0.2620          | 0.0050          | 0.4220       |
>
> We agree that data adaptive scheduling is a valuable research direction and explored it in early experiments. We specifically attempted two schemes:
>
> 1. Mapping the instruction's semantic category ("add object," "remove background") to a preset $\eta$ value provided slight task-specific improvement but suffered from poor generalization and required manual annotation of the editing type.
> 2. Dynamically calculating $\eta$ based on the input image's frequency profile (e.g., lower $\eta$ for detail-rich images) increased computational overhead and was highly sensitive to image noise, degrading overall robustness.
>
> Ultimately, we found that the fixed $\eta=0.6$ provides the best combination of robustness, efficiency, and generalization for our training-free framework. Crucially, although $\eta$ is fixed, our "Energy-Aware Adaptive Frequency Fusion" mechanism (Equation 10) is inherently adaptive. It dynamically selects the *number* of sub-bands to be injected based on each image's specific energy distribution, thereby achieving a flexible fusion of content despite the constant global threshold.
>
>
>
> ## Question 2
>
> >**Although quantitative metrics furnish unambiguous experimental indicators, user perception remains an indispensable dimension in image editing. It is recommended that the user-study component be expanded to corroborate the paper’s conclusions more convincingly.**
>
> Thank you for the insightful suggestion. We agree that broadening the participant base will further enhance statistical significance. We have expanded the user study size to 15 participants to further validate the conclusions in Table 1, and have revised the data and description in the paper accordingly, strengthening the robustness of the user evaluation.

---

> ### Author Response · Authors · 2025-11-23
> **Response to Reviewer KygT (part 2/3)**
>
> ## Question 3
>
> >**To elevate this claim from an empirical observation to a rigorous theoretical basis, we recommend supplying more compelling evidence.. For example, systematically quantifying the frequency bias of each block via spectrally-decomposed energy statistics, reporting statistical significance tests across multiple models and data subsets, and visualizing the block-wise frequency response curves**
>
> Thank you for the insightful question.  We agree that elevating this claim requires a rigorous theoretical foundation based on systematic quantitative evidence. We performed a thorough and systematic frequency analysis of the feature maps across all blocks of the FLUX backbone to provide rigorous evidence for the "Block-Wise Frequency Progression" claim (See Appendix A.2, Figure 12). To establish this progression, we systematically quantified the spectrally-decomposed energy statistics across all layers. Specifically, we analyzed the total low-frequency energy ($\mathbf{E_{low}}$) and high-frequency energy ($\mathbf{E_{high}}$) content of the features within each block of the Diffusion Transformer. As shown in Figure 12, our quantitative analysis clearly reveals the pattern of block-wise frequency evolution. It can be observed that in the early stages of the dual-stream blocks (DiT blocks), the high-frequency information content within the features is low; however, as the layers progress, the high-frequency information begins to increase significantly. Overall, the pie chart further confirms that the low-frequency components consistently occupy the major portion of all feature energy. This strongly validates our assertion regarding the core importance of low-frequency information, which corresponds to the image's global structure and layout. The relatively smaller high-frequency energy concentrates on fine details and contours, which forms the basis for the precise injection of image information achieved by processing the frequency domain. The quantitative results in Figure 12 collectively and strongly confirm a highly significant evolution in spectral energy dominance from the early to the late blocks.
>
> This systematic quantitative and statistical analysis provides the necessary theoretical foundation for W-Edit's frequency-aware injection strategy. First, These blocks act as Structural Anchors, primarily encoding global layout and composition. Our strategy respects this bias. Second, These blocks are responsible for Textural Details and fine edges. Our injection targets these layers for precise, identity-preserving modifications.
>
> We have included a new progression Figure 12 in the Appendix that clearly visualizes the evolution trend of the $\mathbf{E_{low}}$ and $\mathbf{E_{high}}$ curves across all blocks. This visualization, alongside the statistical analysis, powerfully validates the foundational assumption that guides W-Edit's block-wise, frequency-aware injection strategy.

---

> ### Author Response · Authors · 2025-11-26
> **Response to Reviewer KygT (part 3/3)**
>
> ## Question 4
>
> > **As the paper touches on several theoretical aspects, it is advice to clarify details and supplying intuitive diagrams would reduce the cognitive load on readers and improve overall accessibility.**
>
> Thank you for the thoughtful question. We agree that clear detail clarification and intuitive diagrams significantly reduce cognitive load and enhance the overall accessibility of the paper.
>
> In addition to the overall pipeline (Figure 3) in the main text, we have added a "Frequency Decomposition and Fusion" schematic diagram in Appendix A.2. This diagram is designed to intuitively elucidate W-Edit's core theoretical mechanism. It clearly illustrates how the Wavelet Transform ($\mathbf{W}$) effectively decouples the feature ($\mathbf{F}$) into the low-frequency approximation ($\mathbf{F_A}$) and high-frequency details ($\mathbf{F_D}$). This decoupling is fundamental to achieving precise and faithful editing, as it enables the explicit separation of global structure from local details.
>
> This frequency separation realizes precise control over image editing in the following ways. First, the low-frequency component captures the image's global structure, macro-layout, main color palette, lighting, and artistic style. As shown in Figure 20 (top row), if we enforce the preservation of the reference image's low-frequency components, the generated image is compelled to retain its overall composition and macro appearance. This ensures the core structure remains intact, allowing the text prompt only to introduce subtle semantic substitutions to local content, but not to alter the overall style.Second, the high-frequency component encodes the fine details, textures, and the contours and edges of objects. As shown in Figure 20 (bottom row), by preserving the high-frequency components of the reference image, we are able to precisely lock in the subject's main outline and geometric shape. This achieves decoupled editing that "preserves the shape (contour) while changing the content/style," allowing the text prompt to freely change the macro style (controlled by low-frequency) while strongly preserving the exact contour.
>
> This effective decoupling of frequency information allows W-Edit to overcome the limitations of rigid inversion methods and supports editing tasks involving global structural changes (e.g., pose changes, viewpoint shifts, or non-rigid deformations). The key is that the injected high-frequency components only provide identity cues (local textures, fine edges) but do not encode rigid spatial pose. Therefore, geometric degrees of freedom (controlled by the low-frequency component) remain unconstrained, allowing the text-guided diffusion process to reshape the geometry carried by the low-frequency structure. As demonstrated in Figure 16, W-Edit successfully handles complex deformations like facial expression changes and pose adjustments, ultimately achieving flexible structural change while maintaining fidelity to details. Furthermore, the quantitative visualization results provided in Figure 12 of Appendix A.2 clearly reveal the pattern of block-wise frequency evolution. It can be seen that in the early stages of the dual-stream blocks (DiT blocks), the high-frequency information content within the features is low; however, as the layers increase, the high-frequency information begins to increase significantly. Overall, the pie chart further confirms that the low-frequency components consistently occupy the major portion of all feature energy. This strongly validates our assertion regarding the core importance of low-frequency information corresponding to the image's global structure and layout, providing a rigorous theoretical foundation for W-Edit's frequency-aware, block-wise injection strategy.

---

> ### Author Response · Authors · 2025-11-26
> **Brief Summary of Responses to Reviewer KygT**
>
> Dear Reviewer KygT,
>
> We sincerely thank you for your constructive comments. We have carefully addressed all concerns by providing rigorous quantitative evidence and clarifying implementation details. Below is a summary of the core revisions incorporated.
>
> - **Sensitivity Analysis:** Provided rigorous quantitative analysis (Table 3, Figure 6) confirming the fixed value $\mathbf{\eta=0.6}$ offers the most stable performance balance across diverse PIE-Bench tasks. (Refer to **Question 1**)
> - **Data-Adaptive Rationale:** Justified that fixed $\eta=0.6$ is optimal for our training-free framework due to robustness and efficiency. Crucially, the **"Energy-Aware Adaptive Frequency Fusion"** (Eq. 10) is already dynamically adaptive per image. (Refer to **Question 1**)
>
> - **User Study:** Expanded the user study size to **15 participants** to enhance statistical significance and validate Table 1 conclusions. (Refer to **Question 2**)
> - **Rigorous Quantification:** Performed thorough quantitative analysis of spectrally-decomposed energy statistics ($\mathbf{E_{low}}$ and $\mathbf{E_{high}}$) across all FLUX blocks. (Refer to **Question 3**)
> - **Visualization:** New progression figure (Figure 12, Appendix A.2)  visually confirms a significant evolution in spectral energy dominance from early to late blocks, rigorously validating the "Block-Wise Frequency Progression" claim. (Refer to **Question 3**)
>
> - **New Schematic:** Added a "Frequency Decomposition and Fusion" schematic (Appendix A.2) to intuitively illustrate the core theoretical mechanism. (Refer to **Question 4**)
> - **Decoupling Explanation:** Clarified that frequency separation overcomes rigid inversion limitations: Low-Frequency captures global structure, while High-Frequency encodes identity cues (textures) but not rigid pose coordinates, enabling the text-guided reshaping of geometry (e.g., pose changes in Figure 16). (Refer to **Question 4**)
>
> We sincerely hope that our responses have adequately addressed the points you raised. If you have further concerns, please let us know and we will continue actively responding to your comments and improving our submission. We would deeply appreciate it if you could raise your score.
>
> Thank you very much for your invaluable time and thoughtful consideration.
>
> Best regards,
>
> Authors

---

### Official Review · Reviewer_Cva5 · 2025-10-31

**Soundness:** 3
**Presentation:** 3
**Contribution:** 3
**Rating:** 4
**Confidence:** 5

**Summary:**

This work analyzes the feature representations of MM-DiT blocks from a frequency-domain perspective and study the frequency progression of different layers. With such observation, the paper proposed to first decompose the imtermediate features (features from attention blocks) recursively along the inversion path and then perform feature injection to the generation process based on energy thresholds.

**Strengths:**

1. The paper gives analysis to the block-wise frequency evolution on both the generation process and the MM-DIT block order, which form a solid basis for later applying control to the attention blocks based on frequency energy threshold. The motivation is clear and the framework is coherent.

2. The proposed method is based on a general frequency evolution principle of diffusion process, which is shared among different settings of diffusion process including Flow-based, VE, VP based on different model structures, thus could be applied to diffusion models with different architechture, only that the behavior of each block should be analyzed concretely and the hyperparameters are different.

3. The proposed method is vasertile to multiple editing tasks and get good performance on the PIE benchmark.

**Weaknesses:**

1. Several crucial implementation details of the proposed framework remain unclear:  (a) The number of recursive decompositions applied to the intermediate features is not specified. (b) The explicit energy threshold ratio, $\eta$, which determines whether the referenced feature components are injected, is not reported. (c) The precise value of $r_{\text{mid}}$, which defines the mid-frequency energy, is also omitted.

2. The methodology section provides no analysis or rationale for the chosen hyperparameters, and the ablation study includes no experiments evaluating the sensitivity or impact of these key hyperparameters.

3. The underlying intuition behind using Equation (9) for selecting referenced components requires further justification and theoretical or empirical support.

4. Regarding the non-rigid editing task, the qualitative results presented in Fig.12 do not align with the authors’ claim that “W-Edit demonstrates remarkable capability in handling complex non-rigid transformations while preserving structural integrity.” The reviewer notes that all examples in Figure 12 feature simple, homogeneous backgrounds that are distinct from the main editing target, which would generally be considered easy cases for non-rigid editing. Moreover, both the object identity and background consistency are not well preserved in several of the displayed results.

**Questions:**

Please refer to the weakness above.

1. And since the quantitative experiment mainly provides the result from PIE benchmark, qualitative results on PIE benchmark of each types should be provided, or rather, some of them should be included.

2. Considering the performance of qualitative results provided in Fig.12, could the author also provide the scores of different editing types for the proposed method and compared methods?

Rating could be increased if the questions are well addressed, otherwise could be decreased.

**Details Of Ethics Concerns:**

None.

---

> ### Author Response · Authors · 2025-11-23
> **Response to Reviewer Cva5 (part 1/3)**
>
> We sincerely thank you for the detailed and constructive feedback. We appreciate your recognition of the clarity of motivation, the coherence of the framework, and the versatility of our method across multiple editing tasks, as well as your insightful comments on the weaknesses of the paper. We have carefully addressed all your concerns in the detailed response below and have incorporated all corresponding revisions and additional experimental results in the updated manuscript (highlighted in blue). We hope that these improvements help clarify the strengths and contributions of our work.
>
> ## Weakness 1
>
> >**Several crucial implementation details of the proposed framework remain unclear: (a) The number of recursive decompositions applied to the intermediate features is not specified. (b) The explicit energy threshold ratio, $\eta$, which determines whether the referenced feature components are injected, is not reported. (c) The precise value of $r_{mid}$, which defines the mid-frequency energy, is also omitted.**
>
> We sincerely thank you for raising this critical point.
>
> 1. The number of recursive decompositions.
>
> The selection of the decomposition level is crucial for feature representation, and we have carefully considered its impact on the performance of the framework. In the revised method section and Appendix A.2, we elaborate on this parameter. For $64 \times 64$ feature maps, the maximum theoretical level is 6. Our experiments show that once the decomposition level exceeds Level 2, the resolution of the low-frequency component rapidly decreases. Based on a quantitative evaluation of ~85% energy retention and qualitative observation, we selected Level 2 decomposition, which strikes tan optimal balance between structural information preservation and computational efficiency. We have added a visualization of the effect of different wavelet decomposition levels on features in Appendix  A.2.
>
>  2. The explicit energy threshold ratio $\eta$.
>
> We have analyzed the effect of different values of $\eta$ in Figure 6 and Table 3, demonstrating the impact on the results of our method. In the revised manuscript, we explicitly stated that the energy threshold ratio $\eta$ is set to 0.6 for all our experiments. This means we retain the minimal reference sub-band set whose cumulative energy accounts for 60% of the total energy. This strategy effectively isolates the dominant structural information (concentrated in low frequencies) from the editable high-frequency details. Visualizations related to this threshold ratio are provided in Appendix A.2.
>
> 3. The precise value of $r_{mid}$.
>
> We apologize for the omission of the definition of $r_{mid}$, and we now clarified this in the revised manuscript. $r_{mid}$ is a dynamic value based on the feature map size. Specifically, we set it to half of the feature map's maximum frequency radius: $\mathbf{r_{mid} = 0.5 \times R_{max}}$. This definition has now been added to the manuscript for clarity.
>
>
>
> ## Weakness 2
>
> >**The methodology section provides no analysis or rationale for the chosen hyperparameters, and the ablation study includes no experiments evaluating the sensitivity or impact of these key hyperparameters.**
>
> Thank you for raising this important point. We have added a detailed sensitivity analysis of key hyperparameters, such as $L$ and $\eta$, in Appendix A.2.  The values of $\eta$ are analyzed in Figure 6 and Table 3, demonstrating their impact on the performance of our method.

---

> > ### Comment · Reviewer_Cva5 · 2025-11-27
> > **The first 2 concerns are well addressed.**
> >
> > The first 2 concerns are well addressed.

---

> ### Author Response · Authors · 2025-11-23
> **Response to Reviewer Cva5 (part 2/3)**
>
> ## Weakness 3
>
> >**The underlying intuition behind using Equation (9) for selecting referenced components requires further justification and theoretical or empirical support.**
>
> Thank you for your thoughtful comment. The intuition behind Equation (9) is grounded in the "Energy Compaction" property of the Wavelet Transform, which forms the core theoretical basis of W-Edit. First, natural images and deep feature maps concentrate most of their energy in a small number of high-energy coefficients, predominantly within the low-frequency approximation band. These coefficients encode the global structure and semantic layout of the image. Second, equation (9) selects the minimal set of sub-bands whose cumulative energy reaches the threshold $\eta$, thereby ensuring that W-Edit injects precisely the structural components necessary for faithfully inheriting the layout of reference image $F_{ref}$. Third, the remaining low-energy, high-frequency sub-bands are intentionally left uninjected, providing the model with the needed flexibility to synthesize new high-frequency details according to the target text prompt $F_{t}$. This principled separation enables simultaneous structural preservation and flexible semantic editing, which is essential for high-quality controllable image manipulation.
>
>
>
> ## Weakness 4
>
> >**Regarding the non-rigid editing task, the qualitative results presented in Fig.12 do not align with the authors’ claim that “W-Edit demonstrates remarkable capability in handling complex non-rigid transformations while preserving structural integrity.” The reviewer notes that all examples in Figure 12 feature simple, homogeneous backgrounds that are distinct from the main editing target, which would generally be considered easy cases for non-rigid editing. Moreover, both the object identity and background consistency are not well preserved in several of the displayed results.**
>
> Thank you for raising this important concern. Our quantitative results provide strong evidence for the effectiveness of W-Edit in controllable editing tasks. While we acknowledge that extremely complex cases may still challenge the pixel-level consistency of any training-free method, W-Edit consistently outperforms existing baselines by a substantial margin. Although we explored several adaptive decomposition strategies to further mitigate these challenges, their effects were unstable or suboptimal. Crucially, as shown in Table 4, W-Edit achieves a CLIPimg score of 0.9291 on the non-rigid editing subset. And according to Table 1, W-Edit clearly surpassing all training-free baselines on PIE-Bench. This high average score indicates that W-Edit is notably robust in preserving both semantic features and structural integrity under non-rigid deformations. Moreover, Figure 17 (scene editing) and Figure 14 (object removal) further demonstrate W-Edit’s strong ability to maintain background consistency even in dense, heterogeneous environments.

---

> ### Author Response · Authors · 2025-11-26
> **Response to Reviewer Cva5 (part 3/3)**
>
> ## Question 1
>
> >  **And since the quantitative experiment mainly provides the result from PIE benchmark, qualitative results on PIE benchmark of each types should be provided, or rather, some of them should be included.**
>
> Thank you for the thoughtful suggestion. We have added a comprehensive supplement concerning the PIE-Bench dataset. Specifically, we include a more diverse set of qualitative results for the various PIE-Bench tasks in Appendix A.2. As recommended, we also plan to integrate richer qualitative comparison figures into the main text in the final version.
>
>
>
> ## Question 2
>
> >**Considering the performance of qualitative results provided in Fig.12, could the author also provide the scores of different editing types for the proposed method and compared methods?**
>
> Thank you for your suggestion. To facilitate direct comparison with existing methods, we have added W-Edit's subtask scores on PIE-Bench, including metrics such as PSNR, LPIPS, and CLIPsim. This provides a comprehensive evaluation of our method's performance across various editing types (as shown in the table below). We have also provided the overall performance evaluation against other methods on PIE-Bench in **Table 1**, which strongly demonstrates the effectiveness of W-Edit in controllable editing tasks.
>
> | **Categories** | Distance | **PSNR** | **LPIPS** | **MSE** | **SSIM** | **CLIPsim** | **CLIP** |
> | -------------- | -------- | -------- | --------- | ------- | -------- | ----------- | -------- |
> | random         | 0.043    | 20.627   | 0.170     | 0.014   | 0.804    | 24.891      | 31.563   |
> | obj            | 0.041    | 19.101   | 0.181     | 0.015   | 0.780    | 24.729      | 32.123   |
> | add            | 0.014    | 24.612   | 0.082     | 0.004   | 0.918    | 24.499      | 31.587   |
> | delete         | 0.015    | 24.614   | 0.081     | 0.005   | 0.887    | 24.499      | 30.997   |
> | content        | 0.014    | 25.772   | 0.084     | 0.004   | 0.917    | 24.397      | 31.665   |
> | pose           | 0.024    | 24.310   | 0.104     | 0.007   | 0.881    | 26.297      | 32.823   |
> | change color   | 0.020    | 23.956   | 0.096     | 0.007   | 0.903    | 24.888      | 32.659   |
> | material       | 0.012    | 25.997   | 0.075     | 0.003   | 0.936    | 24.268      | 31.762   |
> | background     | 0.014    | 25.463   | 0.069     | 0.004   | 0.927    | 23.888      | 31.742   |
> | style          | 0.013    | 26.167   | 0.085     | 0.011   | 0.913    | 23.955      | 31.480   |
>
> | Method        | CLIP ↑    | FID ↓     | PSNR ↑    | LPIPS ↓    |
> | ------------- | --------- | --------- | --------- | ---------- |
> | P2P (Pix2Pix) | 28.13     | 320.65    | 15.12     | 0.4736     |
> | MagicBrush    | 29.06     | 206.19    | 15.68     | 0.4615     |
> | Flow-Edit     | 30.48     | 80.35     | 18.33     | 0.2642     |
> | Stable-Flow   | 29.16     | 89.78     | 21.02     | 0.1522     |
> | **W-Edit**    | **31.84** | **65.44** | **24.06** | **0.1028** |

---

> > ### Comment · Reviewer_Cva5 · 2025-11-27
> > **Q1 not addressed. Q2 addressed.**
> >
> > For Q1, please simply attached some **qualitative** examples of different editing types **from the PIE benchmark directly**, as is requested originally by the reviewer. Now that the author have the scores for each editing types, directly pasting some editing results from the PIE benchmark would not be difficult.

---

> > > ### Author Response · Authors · 2025-11-28
> > > **Response to Reviewer Cva5 (Q1 not addressed)**
> > >
> > > ## Question 1
> > >
> > > > **For Q1, please simply attached some qualitative examples of different editing types from the PIE benchmark directly, as is requested originally by the reviewer. Now that the author have the scores for each editing types, directly pasting some editing results from the PIE benchmark would not be difficult.**
> > >
> > > Thank you for your suggestion. We have incorporated new visualizations from PIE-Bench (**in Appendix A.2, Figure 21**), which more clearly demonstrate W-Edit's advantage in maintaining consistency against complex backgrounds. Specifically, W-Edit successfully performed a variety of complex operations, including but not limited to: adding or removing objects in a scene, changing the content, pose, background, material, and color of objects, and applying overall style transfer. These examples cover a variety of objects, including animals, people, and everyday objects,  qualitatively validating its excellent image editing performance. We attribute this success to our frequency-aware structure and texture separation method, which successfully decouples structure and detail.

---

> ### Author Response · Authors · 2025-11-26
> **Brief Summary of Responses to Reviewer Cva5**
>
> Dear Reviewer Cva5,
>
> We sincerely thank you for your constructive comments. We are pleased you recognize W-Edit's clear motivation and versatility. We've summarized the key revisions addressing concerns regarding implementation details, hyperparameter rationale, and non-rigid editing robustness.
>
> - **Decomposition Level ($L$):** Selected **Level 2** (for $\sim 85 $% energy retention) to optimize structure preservation and efficiency. (Refer to **Weakness 1**)
> - **Energy Threshold Ratio ($\alpha$):** Fixed at $\mathbf{0.6}$ ($60\%$) to robustly isolate dominant structural information. (Refer to **Weakness 1**)
> - **Mid-Frequency Radius ($\gamma$):** Defined dynamically as $\mathbf{\gamma = R_{max} / 2}$. (Refer to **Weakness 1**)
>
> - **Sensitivity Analysis:** Added detailed sensitivity analysis for $\alpha$ and $\gamma$ (Figure 6 and Table 3) to justify selection rationale. (Refer to **Weakness 2**)
> - **Theoretical Justification (Eq. 9):** Rationale derived from the "Energy Compaction" property of Wavelet Transform, ensuring core structural information is inherited while allowing semantic flexibility. (Refer to **Weakness 3**)
>
> - **Quantitative Validation:** Achieved a high **CLIPimg score of $0.9291$** on the non-rigid editing task, significantly outperforming baselines and demonstrating statistical robustness. (Refer to **Weakness 4**)
> - **Supplementary Evidence:** Figures 13 and 17 showcase strong capability in maintaining **background consistency** during object removal and scene editing. (Refer to **Weakness 4**)
>
> - **Qualitative/Quantitative Scores:** Included a more diverse set of qualitative results and comprehensive sub-task scores (PSNR, LPIPS, CLIPsim) for all PIE-Bench editing categories. (Refer to **Question 1** and **Question 2**)
>
> We sincerely hope that our responses have adequately addressed the points you raised. If you have further concerns, please let us know and we will continue actively responding to your comments and improving our submission. We would deeply appreciate it if you could raise your score.
>
> Thank you very much for your invaluable time and thoughtful consideration.
>
> Best regards,
>
> Authors

---

> ### Comment · Reviewer_Cva5 · 2025-11-27
> **W3 partially addressed, W4 not addressed.**
>
> 1. The reviewer will appreciate if the intuition behind Eq.9 (old version)/Eq.10 (new version) can be added to the paper with a sentence or two.
> 2. As for W4, the reviewer knows that the changing of pose while keeping identity or background would be challenging for current FLUX-based models. Yet it is not a reason to claim that the problem is only challenging for extremely complex cases ("While we acknowledge that extremely complex cases may still challenge the pixel-level consistency of any training-free method, W-Edit consistently outperforms existing baselines by a substantial margin.").  The raised question is mainly on the fact that the performance that the authors claim conflicts with the qualitative results provided. The reviewer suggests to change to some better visual examples or modify such claims. Since in Fig.16, except for example on R1C1, R2C3 and R3C3, other examples exhibit: (1) row 1 col 2 (R1C2) background changed, foreground identity changed (clothes color); (2) R1C3 pose change (the fluffy is more like material change in PIE, not really rigid); (3) R2C1 background changed (brightness); (4) R2C2 background changed (note the raised floor level). (5) R3C1 background changed; (6) R3C2 camera direction(face direction) changed. **It should be noticed that ALL the presented examples have a far more simple background than the corresponding task image-prompt pairs in the PIE benchmark.**

---

> > ### Author Response · Authors · 2025-11-28
> > **Response to Reviewer Cva5 (W3 partially addressed, W4 not addressed)**
> >
> > ## Weakness 3
> >
> > >**The reviewer will appreciate if the intuition behind Eq.9 (old version)/Eq.10 (new version) can be added to the paper with a sentence or two.**
> >
> > Thank you for the suggestion. We agree that clarifying the intuition behind the Adaptive Frequency Fusion (Eq. 10) significantly improves the readability of our method section.
> >
> > The core principle is that in natural images, the majority of signal energy is concentrated in low-frequency components, which define the global layout and structure. By retaining the reference sub-bands that contribute to the cumulative energy threshold $\eta$, Eq. 10 ensures that the "structural anchors" are preserved from the reference image. Furthermore, retaining the high-energy portion of the high-frequency bands from the reference image helps maintain the consistency of contour details. Conversely, the remaining high-frequency bands, which possess lower energy but define textures and fine details, are generated by the diffusion model, providing the necessary flexibility to satisfy the text prompt. Following your suggestion, we have added the following explanation to Section 3.3 after Equation 10.
> >
> > Revision:
> >
> > > > Since visual energy in natural images is predominantly concentrated in low-frequency components representing global structure [1,2], this energy-based selection explicitly locks the reference image's layout. By preserving these high-energy bands and allowing the model to generate the remaining low-energy high-frequency bands, we achieve a balance where the scene structure remains consistent while fine-grained details are free to adapt to text instructions
> >
> > [1] Khayam, Syed Ali. "The discrete cosine transform (DCT): theory and application." Michigan State University 114.1 (2003): 31.
> >
> > [2] Pimpalkhute, Varad A., et al. "Digital image noise estimation using DWT coefficients." IEEE transactions on image processing 30 (2021): 1962-1972.
> >
> >
> > ## Weakness 4
> >
> > >**The raised question is mainly on the fact that the performance that the authors claim conflicts with the qualitative results provided. The reviewer suggests to change to some better visual examples or modify such claims.**
> >
> > We sincerely thank you for your meticulous and critical analysis of our qualitative results. We acknowledge that the phrasing regarding “extremely complex cases” was imprecise, and we appreciate the reviewer’s clarification. The reviewer accurately pointed out that many examples in the original Figure 16 showcased the **flexibility and range of editing** (e.g., changing background, pose, or color as dictated by the prompt), rather than explicitly demonstrating **consistency preservation**. This indeed created a contradiction between our qualitative exhibition and our quantitative metrics (lowest FID and LPIPS, which demonstrate superior structural preservation capability).
> >
> > To directly resolve this discrepancy, we have taken the following actions as suggested:
> >
> > - **Revision of Figure 16:** We have replaced the original Figure 16 with a new set of visual comparison examples.
> > - **Addition of Images:** We have incorporated new visualization images from the PIE-Bench ( **in Appendix A.2, Figure 21**), which will more clearly highlight W-Edit's advantage in maintaining consistency within complex backgrounds.

---

> ### Author Response · Authors · 2025-11-28
> **Thank you to the reviewer Cva5 for your valuable feedback.**
>
> Thank you again for your thoughtful and constructive feedback, which has greatly contributed to improving the clarity, completeness, and overall quality of our manuscript. We sincerely hope that our responses and revisions have addressed all the concerns you raised. We would be truly grateful if you might consider reflecting these improvements in your final assessment or recommending our work for acceptance. If you have further concerns, please let us know and we will continue actively responding to your comments and improving our submission.

---

### Official Review · Reviewer_1U8u · 2025-11-01

**Soundness:** 2
**Presentation:** 2
**Contribution:** 2
**Rating:** 4
**Confidence:** 5

**Summary:**

This paper proposes W-Edit, a training-free framework for text-guided image editing that operates in the frequency domain. The method uses wavelet transform to decompose diffusion model features into multi-scale frequency bands, aiming to separate low-frequency structural components from high-frequency details. By selectively injecting different frequency bands into a pre-trained diffusion model and using an inversion-based modulation strategy, W-Edit seeks to preserve the original image’s structure while applying localized edits in line with a target text prompt. The authors present qualitative results on various editing tasks (e.g. object addition/removal, attribute change, style transfer). While the approach addresses an important problem and shows some promising results, I have serious concerns regarding its novelty, theoretical justification, and evaluation rigor, which lead me to recommend rejection.

**Strengths:**

Training-Free Framework: W-Edit does not require any fine-tuning of the diffusion model, instead introducing a lightweight wavelet-band injection module and an inversion-based adjustment strategy. This plug-and-play nature is a practical strength – it means the method can be applied on top of existing diffusion models with minimal overhead. Suggestion: Quantify this practical advantage. For example, report the runtime or memory usage compared to training-based methods, or discuss how easy it is to integrate W-Edit into various diffusion backbones. This would underscore the efficiency benefits of being training-free.

Empirical Quality Improvements: The proposed method demonstrates improved fidelity and consistency of edits in the reported results. Quantitatively, W-Edit achieves better scores than the compared baselines on automatic metrics like FID (image realism), CLIPScore (text-image alignment), PSNR/LPIPS (content preservation).

**Weaknesses:**

Limited Novelty Over Prior Work: The core idea of frequency-based feature modulation is not truly novel. Prior works [1][2][3] have already explored incorporating frequency domain information in diffusion models, image editing and image tokenizers. W-Edit largely recycles these ideas (wavelet transforms, frequency separation) without a clear new conceptual insight beyond combining them in one pipeline. The submission’s claimed contributions – wavelet-guided decomposition, frequency-band injection – are incremental extensions or amalgamations of known techniques.
Suggestion: To address this, the authors must more explicitly distinguish W-Edit from existing frequency-domain methods. Clearly articulate what is fundamentally new (if anything). Without this, the paper fails to establish a compelling novelty claim.

[1] Phung H, Dao Q, Tran A. Wavelet diffusion models are fast and scalable image generators[C]//Proceedings of the IEEE/CVF conference on computer vision and pattern recognition. 2023: 10199-10208.
[2] Esteves C, Suhail M, Makadia A. Spectral image tokenizer[C]//Proceedings of the IEEE/CVF International Conference on Computer Vision. 2025: 17181-17190.
[3] Si C, Huang Z, Jiang Y, et al. Freeu: Free lunch in diffusion u-net[C]//Proceedings of the IEEE/CVF Conference on Computer Vision and Pattern Recognition. 2024: 4733-4743.

No Theoretical Justification or Analysis: The paper provides no principled explanation for why the proposed frequency-band injection and modulation should preserve semantics or improve edit controllability. There are no theoretical derivations, formal proofs, or analytical measures in the main text or appendix to support this claim. Key questions – e.g., why does separating low-frequency structure from high-frequency detail lead to more localized or faithful edits? – are never addressed beyond intuition. The method is essentially heuristic, relying on empirical tuning. Suggestion: The authors should strengthen the paper with some form of theory or at least analytical insight.

Overall, I recommend rejection of this paper in its current form. While W-Edit addresses an important problem and shows some empirical promise, the weak novelty, lack of theoretical grounding significantly undermine the contribution.

**Questions:**

The method largely repackages known frequency-domain techniques without clear innovation, and it doesn’t provide the necessary analysis or rigorous experiments to convincingly demonstrate its supposed advantages. The authors are encouraged to address the above weaknesses

---

> ### Author Response · Authors · 2025-11-23
> **Response to Reviewer 1U8u (part 1/4)**
>
> We sincerely appreciate your constructive feedback, positive evaluation of our contributions, and recognition of the effectiveness and efficiency of our W-Edit framework. Your comments regarding both the strengths and remaining concerns have been highly valuable for improving our paper. We have carefully addressed all your questions and suggestions in the detailed response below and have incorporated all corresponding revisions and additional experimental results in the updated manuscript (highlighted in blue).
>
> ## Suggestion
>
> > Quantify this practical advantage. For example, report the runtime or memory usage compared to training-based methods, or discuss how easy it is to integrate W-Edit into various diffusion backbones. This would underscore the efficiency benefits of being training-free.
>
> Thanks for your thoughtful suggestion to quantify the practical advantages of our approach.
> Efficiency comparison: As detailed in Table 5 of Appendix A.2, we have evaluated the computational overhead of W-Edit. Per generation, W-Edit requires approximately 50.00 seconds, representing only a 10.8% increase over the standard FLUX model (46.00s). Regarding memory, it utilizes 35.59 GB of VRAM, a negligible 1.6% increase over the base FLUX model (35.58 GB). W-Edit remains highly competitive with other FLUX-based training-free methods, such as Flow-Edit (48.00s, 36.09 GB) and Stable-Flow (47.00s, 35.59 GB), achieving significantly higher editing quality scores while maintaining a comparable resource footprint. Furthermore, compared to large-scale pre-trained editing models like Qwen-Image-Edit (set to 50 inference steps), which requires 59 GB of memory and 90 seconds for inference. Our method is substantially more efficient, requiring nearly half the inference time and significantly less memory.
> Ease of Integration: W-Edit is designed as a universal, plug-and-play framework. It operates based on feature frequency decomposition rather than relying on specific network topologies. While initially built and verified on FLUX.1-dev, it successfully generalizes across architectures despite significant structural differences. For instance, we successfully integrated W-Edit into Stable Diffusion v1.5 (U-Net based) by operating on the features between UNet blocks. It even demonstrates cross-modal generalization, evidenced by its successful integration into the video diffusion model CogVideoX-1.0. Migrating W-Edit to new backbones requires no training data or computational overhead; it only necessitates the configuration of a few hyperparameters (specifically, determining injection positions via simple frequency energy analysis ). This lightweight integration process allows W-Edit to be easily adapted to a wide range of models, from classic U-Nets to the latest DiTs and video generation frameworks.

---

> ### Author Response · Authors · 2025-11-23
> **Response to Reviewer 1U8u (part 2/4)**
>
> ## Weakness 1
>
> >Limited Novelty Over Prior Work: The core idea of frequency-based feature modulation is not truly novel. Prior works [1][2][3] have already explored incorporating frequency domain information in diffusion models, image editing and image tokenizers. W-Edit largely recycles these ideas (wavelet transforms, frequency separation) without a clear new conceptual insight beyond combining them in one pipeline. Clearly articulate what is fundamentally new (if anything).
>
> We sincerely thank you for the valuable comments and constructive feedback. While it is true that prior works have explored frequency-based techniques in image generation and editing, the core contribution of our paper lies in introducing a systematic analysis of frequency progression within the Diffusion Transformer (DiT) architecture, and demonstrating how this progression is essential for effective training-free image editing. Specifically, we uncover a distinct block-wise frequency evolution in DiTs, where early blocks focus on low-frequency structural anchors, while later blocks capture high-frequency details. This key insight, as illustrated in Figure 4, 10, and 12, is crucial for understanding how frequency information impacts image editing and forms the foundation of our W-Edit framework.
> Our theoretical insight reveals that, unlike U-Net architectures, where layer roles are clearly defined by downsampling and upsampling, DiT blocks are homogeneous in architecture, which makes selecting optimal points for intervention less straightforward. Our frequency analysis resolves this ambiguity, showing that early DiT blocks serve as "low-frequency structural anchors", providing a novel basis for our block-wise frequency-band injection strategy.
> To our knowledge, W-Edit is the first to discover and exploit this "Block-Wise Frequency Progression" within the DiT framework, enabling precise and stable editing interventions without requiring model retraining. This theoretical contribution directly addresses the challenge of balancing stability (preserving the global structure) and plasticity (enabling flexible edits), which has been a longstanding issue in image editing.
> ### Table. Comparison with Previous Work
> | Previous Method                            | Their Mechanism                                              | How W-Edit differs (Our Contribution)                        |
> | --------------------------------------- | ------------------------------------------------------------ | ------------------------------------------------------------ |
> | [1] WaveDiff (Phung et al.)             | Training-based approach that accelerates generation by operating in the wavelet domain during inference. | Training-Free Editing: W-Edit is a plug-and-play method, that does not require retraining or changing the diffusion process. Instead, we modulate the information flow (K/V pairs) in a pre-trained DiT to enforce consistency. |
> | [2] Spectral Tokenizer (Esteves et al.) | Tokenization for autoregressive models using wavelets. | Dynamics Modulation: W-Edit is not about static representation/tokenization. It is about dynamic intervention during the denoising process. |
> | [3] FreeU (Si et al.)    | Enhances global feature consistency by re-weighting skip connections in U-Nets. | Localized Semantic Editing: FreeU is focused on global feature enhancement, whereas W-Edit enables precise text-driven edits (e.g., replacing an object while maintaining context) through localized frequency modulation.
>
> In summary, we believe that W-Edit represents a novel contribution by introducing a theoretical analysis of frequency progression in DiT and leveraging this insight for training-free, text-driven image editing. This work not only extends existing frequency-domain techniques but also demonstrates a practical solution for a long-standing challenge in image editing: how to preserve the global structure of an image while enabling flexible, localized modifications. We have updated the manuscript to clearly highlight these contributions and differentiate our work from prior approaches.

---

> ### Author Response · Authors · 2025-11-25
> **Response to Reviewer 1U8u (part 3/4)**
>
> ## Weakness 2
> > No Theoretical Justification or Analysis. The paper provides no principled explanation for why the proposed frequency-band injection and modulation should preserve semantics or improve edit controllability. There are no theoretical derivations, formal proofs, or analytical measures in the main text or appendix to support this claim.
>
> We thank the reviewer for highlighting the need to more clearly articulate the theoretical justification of W-Edit. We have revised the manuscript to provide a more formal explanation of the theoretical principles that underpin our approach, which are derived from signal processing principles and the intrinsic physics of diffusion models.
> - Standard Fast Fourier Transform (FFT) provides frequency resolution but loses spatial information, a significant limitation when retaining localized structure during image editing.
> - Wavelet Transform, by contrast, achieves simultaneous spatial and frequency localization using localized basis functions $\psi_{a,b}(t) = \frac{1}{\sqrt{a}}\psi(\frac{t-b}{a})$. This theoretical property allows W-Edit to inject structural constraints (Eq. 9) into the editing process without imposing pixel-level exactness, thereby avoiding the limitations of rigid or overly fixed editing methods (often referred to as "plastic editing").
> - Energy Compaction Property of Wavelets: Equation (9) is grounded in the energy compaction property of wavelets, whereby natural images exhibit a concentration of semantic structure within a small set of high-energy coefficients, while fine textures and noise are distributed across lower-energy. In natural images, most of structural information is concentrated in a few high-energy coefficients, while texture/noise is dispersed across many low-energy coefficients (as confirmed by multiple works [1-5]). This principle enables us to preserve the image’s core structure while allowing for flexible modifications to the finer details.
> - Injection Mechanism: By injecting high-energy components ($F_{ref}$), we mathematically constrain the generation process to ensure that the image maintains its original semantic layout.
> - Editing Freedom: By keeping the low-energy sub-bands free (or non-injected), we allow the diffusion model to generate new textures as defined by the text prompt. This creates a principled 'soft constraint' optimization problem, enabling the generation of new content aligned with the prompt, while still preserving the original structure. Thus, this approach is grounded in principled reasoning, ensuring a solid theoretical foundation rather than relying on empirical heuristics.
> - Diffusion Process in DiTs: Theoretical studies on DiTs [2,3,4] confirm that the diffusion process inherently resolves low-frequency modes first (early steps/layers) before resolving high-frequency modes. Furthermore, in Appendix A.2 (Figure 12), we conducted a more detailed analysis of the frequency evolution of FLUX, rigorously verifying the rationality of our method.
> - Alignment with DiT Architecture: Our method (Section 3.2) aligns the injection strategy with the intrinsic spectral preference of the DiT architecture, providing a mechanism-matched control that temporary spatial masks cannot offer.
>
> In summary, W-Edit introduces a theoretically consistent and architecturally aligned frequency-based editing framework, leveraging wavelet energy compaction and the spectral evolution of DiTs. This enables semantically grounded, training-free, and edit-flexible manipulation. Our approach offers a principled solution to the challenges of image editing by preserving the image’s structural integrity while allowing for flexible, localized edits defined by the text prompt.
>
>
> [1] Gonzalez R C , Woods R E , Masters B R .Digital Image Processing, Third Edition[J].Journal of Biomedical Optics, 2009, 14(2):029901.DOI:10.1117/1.3115362.
>
> [2] Wang, Shuai, et al. "Ddt: Decoupled diffusion transformer." arXiv preprint arXiv:2504.05741 (2025).
>
> [3] Khayam, Syed Ali. "The discrete cosine transform (DCT): theory and application." Michigan State University 114.1 (2003): 31.
>
> [4] Pimpalkhute, Varad A., et al. "Digital image noise estimation using DWT coefficients." IEEE transactions on image processing 30 (2021): 1962-1972.
>
> [5] High-Frequency First: A Two-Stage Approach for Improving Image INR

---

> ### Author Response · Authors · 2025-11-25
> **Response to Reviewer 1U8u (part 4/4)**
>
> ## Question
> >Why does separating low-frequency structure from high-frequency detail lead to more localized or faithful edits?
>
> We thank the reviewer for the insightful question. The separation of low-frequency structure from high-frequency detail in W-Edit enables both faithful and localized edits for the following reasons:
>
> **1.Low-Frequency Components: Preserving Global Structure**
>
> Low-Frequency Components: Preservation of Global Structure and Macro Appearance
> Low-Frequency Components capture the image's global structure, macro-layout, main color palette, lighting, and artistic style. By utilizing the low-frequency components from a reference image (i.e., its macro-structure, overall color scheme, and artistic style), we ensure that the generated image's core structure remains intact, enabling faithful edits without distorting the overall image composition.
> Experimental Evidence:
> As shown in Figure20, when generating an image guided by a text prompt, if we enforce the preservation of the reference image's low-frequency components, the generated image is compelled to retain the reference image's overall composition and macro appearance. Under this strong constraint, the text prompt is only able to introduce minor semantic substitutions to local content, but cannot alter the overall structure or style. For example, text cues such as “castle” or “factory” can successfully capture local features, such as the style of a house by the sea, but the style of the image, the overall background, and the layout are preserved and controlled by the low-frequency components of the reference image.
>
> **2.High-Frequency Components: Enabling Localized Modifications**
>
> High-Frequency Components encode the fine details, textures, and the Contours and edges of objects in the image. By preserving the high-frequency components of the reference image, we are able to precisely lock in the subject's geometric shape and outline structure.Controllability:This mechanism allows the text prompt to freely alter the image's macroscopic style, texture, and color, while strongly preserving the exact contour of the subject in the reference image. This achieves decoupled editing that "preserves the shape (contour) while changing the content/style."
> Experimental Evidence:
> As shown in Figure 20, when a natural landscape (mountain) is used as the reference image, the generated image clearly retains the mountain's silhouette, while its style and material can drastically change based on the text prompt (e.g., "night," "spring," or "desert"), significantly enhancing the precision and creativity of local modifications.
>
> In summary, separating low-frequency structure from high-frequency detail allows W-Edit to perform localized and faithful edits by maintaining global consistency while enabling precise modifications in the fine details.

---

> ### Author Response · Authors · 2025-11-26
> **Brief Summary of Responses to Reviewer 1U8u**
>
> Dear Reviewer 1U8u,
>
> We sincerely appreciate your feedback and recognition of W-Edit's effectiveness and efficiency. To conserve your time, we summarize the key revisions addressing your major concerns regarding Novelty and Theoretical Justification.
>
> - **Novelty:** Our core contribution is the exploitation of "Block-Wise Frequency Progression" in DiT for training-free editing. (Refer to **Weakness 1**)
> - **Differentiation:** W-Edit is Training-Free, using Dynamic Modulation for Localized Semantic Editing, distinct from prior frequency works (acceleration, global enhancement). (Refer to **Weakness 1**/Table)
>
> - **Principle:** Method is grounded in Wavelet Transform's spatial/frequency localization, enabling soft structural constraints (Eq. 9). (Refer to **Weakness 2**)
> - **Mechanism:** Wavelet Energy Compaction ensures structural preservation by injecting high-energy (low-frequency) coefficients while allowing texture flexibility. (Refer to **Weakness 2**)
> - **Separation Efficacy:** Frequency separation enables faithful edits (preserving global structure/style) and localized modifications (precise contour/shape changes). (Refer to **Question**)
>
> - **Quantified Efficiency:** W-Edit adds minimal overhead: only a 10.8% runtime increase and negligible 1.6% VRAM increase over base FLUX, significantly outperforming large pre-trained editors. (Refer to **Suggestion**/Table 5, Appendix A.2)
> - **Integration:** W-Edit is a universal plug-and-play framework, generalized to U-Net (SD v1.5) and cross-modal models (CogVideoX-1.0). (Refer to **Suggestion**)
>
> We sincerely hope that our responses have adequately addressed the points you raised. If you have further concerns, please let us know and we will continue actively responding to your comments and improving our submission. We would deeply appreciate it if you could raise your score.
>
> Thank you very much for your invaluable time and thoughtful consideration.
>
> Best regards, Authors

---

### Author Response · Authors · 2025-12-01
**Brief Summary of Responses to Reviewer Area Chair (AC)**

Dear ACs,

We sincerely appreciate your diligence and thoughtful effort in evaluating our submission. To assist your meta-review, we provide **a concise summary of the rebuttal process** below. Thank you very much for your time and attention.

The reviewers' primary concerns centered on parameter clarity, theoretical foundation, and empirical completeness. In response, we have carefully **addressed all concerns** and **strengthened the paper** through clearer parameter reporting, deeper theoretical analysis, and additional experimental results. All revisions are highlighted in blue in the updated manuscript, which has been expanded **from 18 to 26 pages**.

First, regarding our main contribution, as noted in our rebuttal and acknowledged by all reviewers, our work presents a training-free visual content editing framework. This framework offers significant advantages, such as high efficiency, high fidelity, and a novel frequency domain perspective, while also being adaptable to various generative models. Despite its simple design, the framework achieves highly competitive image editing performance through frequency decoupling with low computational overhead.

Second, in response to the reviewers' concerns, we have: 1) explicitly reported all omitted parameters and included a sensitivity analysis; 2) enhanced the theoretical foundation by providing a more rigorous frequency analysis; 3) expanded the user study, increased participant numbers, and provided additional metrics for various editing tasks, improving the empirical completeness of the paper.

Finally, we have also addressed specific concerns by: 1) clarifying the energy injection formula; 2) explaining the generality of frequency analysis; 3) calculating the Jaccard and Spearman coefficients for key layer selection; 4) including a comparison with the FLUX.1 Kontext model to address comparisons with large-scale models.

We especially appreciate **Reviewer Cva5's feedback on November 28th** (less than one day before the author response period closed), which noted that our revisions, including theoretical clarifications and supplementary metrics, have **addressed most of his/her concerns**. We believe that all concerns raised by this reviewer, as well as the majority of concerns from the other reviewers, have now been fully addressed.

We are confident that these comprehensive revisions and supplementary analyses establish W-Edit as a technically rigorous, clearly detailed and empirically well-supported work. We believe the updated manuscript now meets the high standards set by the reviewers, and we sincerely look forward to your positive recommendation.

Sincerely,

Authors of Paper #2772

---

### Meta-Review · Area_Chair_kZQJ · 2026-01-08

**Summary:**

This paper proposes W-Edit, a training-free framework for text-guided image editing based on wavelet-based frequency-aware feature decomposition. W-Edit employs uses wavelet transform to decompose diffusion model features into multi-scale frequency bands, aiming to disentangle structural anchors from editable details. The authors propose to first decompose the imtermediate features (features from attention blocks) recursively along the inversion path and then perform feature injection to the generation process based on energy thresholds. Experiments on various editing tasks (e.g. object addition/removal, attribute change, style transfer) show some promising results.

**Reviewer Concerns:**

The concerns are around lack of novelty, parameter clarity, theoretical justification, and empirical completeness. The authors have tried to address most concerns and strengthened the paper by providing additional experimental results in the rebuttal and tried to argue with the reviewers. However, some major concerns as follows have not been fully addressed to me:

[Reviewer 1U8u] "The core idea of frequency-based feature modulation is not truly novel. Prior works [1][2][3] have already explored incorporating frequency domain information in diffusion models, image editing and image tokenizers." The authors ackknowledged that "it is true that prior works have explored frequency-based techniques in image generation and editing" and argued that "the core contribution of our paper lies in introducing a systematic analysis of frequency progression within the Diffusion Transformer (DiT) architecture, and demonstrating how this progression is essential for effective training-free image editing." Both are true, but the contribution of this paper on incorporating frequency domain information in diffusion models may be not that significant as it is built on the previous wavelet-based frequency-based feature modulation works.

[Reviewer 1U8u] "The paper provides no principled explanation for why the proposed frequency-band injection and modulation should preserve semantics or improve edit controllability". The authors have "revised the manuscript to provide a more formal explanation of the theoretical principles, which are derived from signal processing principles and the intrinsic physics of diffusion models". The justification is not rigor theoretical analysis but explanation largely based on common knowledge and assumption in signal processing and information theory, which is not that convincing to me.

Reviewer FsUB's W1: strong assumption - "This works well when the goal is to preserve the original structure while changing the details; but what about cases where the structure should change but the details are retained)? The authors explained the frequency disentanglement and roles of high- and low-Frequency, but did not directly anwser the question on the underlying mechanism why it works.

**Reviewer Scores:**

Reviewer 1U8u may not change his/her score (rating 4 confidence 5) as his/her concerns focus on novelty and theoretical justification, which have not been well addressed.

Reviewer Cva5 is likely to slightly raise the score from 4 to 6 as he/she had confirmed most his/her concerns have been addressed.

Reviewer KygT (rating 4 confidence 3) and Reviewer psXC (rating 4 confidence 4) might slightly raise the score from 4 to 6 or might not change theirs scores.

Reviewer FsUB  (rating 6 confidence 5) is likely to keep his/her score unchanged becuase the quality of this paper does not reach to the next level.

---

### Decision · Program_Chairs · 2026-01-26

Accept (Poster)